# On the Role of Depth and Looping for In-Context Learning with Task Diversity

## Abstract

The intriguing in-context learning (ICL) abilities of *deep Transformer models* have lately garnered significant attention. By studying in-context linear regression on unimodal Gaussian data, recent empirical and theoretical works have argued that ICL emerges from Transformers' abilities to simulate learning algorithms like gradient descent. However, these works fail to capture the remarkable ability of Transformers to learn *multiple tasks* in context. To this end, we study in-context learning for linear regression with diverse tasks, characterized by data covariance matrices with condition numbers ranging from $[1, \kappa]$, and highlight the importance of depth in this setting. More specifically, (a) we show theoretical lower bounds of $\log(\kappa)$ (or $\sqrt{\kappa}$) linear attention layers in the unrestricted (or restricted) attention setting and, (b) we show that *multilayer Transformers* can indeed solve such tasks with a number of layers that matches the lower bounds. However, we show that this expressivity of multilayer Transformer comes at the price of robustness. In particular, multilayer Transformers are not robust to even distributional shifts as small as $O(e^{-L})$ in Wasserstein distance, where $L$ is the depth of the network. We then demonstrate that Looped Transformers —a special class of multilayer Transformers with weight-sharing— not only exhibit similar expressive power but are also provably robust under mild assumptions. Besides out-of-distribution generalization, we also show that Looped Transformers are the only models that exhibit a monotonic behavior of loss with respect to depth.

## 1 Introduction

Transformer-based language models (Vaswani, 2017) have demonstrated remarkable in-context learning abilities (Brown et al., 2020), thereby bypassing the need to fine-tune them for specific tasks. This is a desirable feature for large models because fine-tuning them is often expensive. In particular, given just a few samples of a new learning task presented in the context, the model is able to meta-learn the task and generate accurate predictions without having to update its parameters. In an attempt to study this phenomenon, recent work has shown that Transformers can demonstrate such in-context learning abilities on a variety of learning tasks such as linear or logistic regression, decision trees (von Oswald et al.; Garg et al., 2022; Xie et al., 2021) and, perhaps surprisingly, even training Transformers themselves (Panigrahi et al., 2023). This has sparked significant interest to theoretically understand the in-context learning phenomenon. From a more theoretical perspective, there are two crucial aspects to this in-context learning ability: (1) the Transformer architecture is powerful enough to implement iterative algorithms, including first order optimization methods such as gradient or preconditioned gradient descent (Li et al., 2024; Von Oswald et al., 2023; Akyürek et al., 2022), and (2) Transformers with appropriate optimization algorithm can indeed learn to simulate such algorithms (Ahn et al., 2024a; Gatmiry et al., 2024). These results provide further insights into how in-context learning could arise within Transformer models. Recent papers (e.g (Ahn et al., 2024a; Gatmiry et al., 2024)) have studied both these aspects for Transformers on a *single task* — linear regression setting where the data is sampled i.i.d Gaussian.

One of the mysterious features of recent foundation models, which separates them from traditional learning systems, is their ability to handle a spectrum of tasks, from question answering within a broad range of topics, to reasoning or in-context learning for various learning setups. The emer-

gence of this ability to handle task diversity naturally necessitates the training data to incorporate a diverse datasets, which inherently makes the training distribution highly multimodal. Consequently, foundation models need more capacity to be able to solve such diverse tasks. However, recent work on in-context learning theory focus on unimodal Gaussian data distribution. In such settings, even constant number of layers suffice to achieve any arbitrarily small error. Thus, while interesting, these works fail to capture the remarkable ability of Transformers to learn *multiple tasks* in context and explicate the role of depth in these settings.

To this end, in this paper we introduce a new in-context learning setting, *task-diverse linear regression*, which specifically examines how well Transformers can in-context learn linear regression problems arising from fairly diverse data. More specifically, in task-diverse linear regression, the data can be sampled from multiple Gaussians where the eigenvalues of its covariance matrix can range from $[1, \kappa]$. For this setting, we ask the following important question:

*For multilayer Transformer, what is the role of depth in task-diverse linear regression? Is it possible to establish upper and lower bounds that depend on task diversity?*

In this paper, we resolve this problem by showing lower bound of $\log \kappa$ and matching upper bounds — indicating a fundamental limit on the depth required to solve the task-diverse linear regression setting. Furthermore, in the special case of attention layers considered in (Von Oswald et al., 2023), we obtain a lower bound of $\Omega(\sqrt{\kappa})$ instead. Correspondingly, we also show a matching upper bound in this special setting, thereby; providing a comprehensive picture of the representation power of multilayer Transformers in various important settings with task diversity.

While the above results paint a powerful picture of multilayer Transformers, it remains unclear if this comes at any cost. In this paper, we show that this is indeed the case and the price we pay is *robustness*. First, diverging significantly from prior works, we study the problem of *out-of-distribution generalization* on a very general class of distributions (see 2). Under this general class of distributions, we show that the out-of-distribution generalization of multilayer Transformer can get exponentially worse with depth. Thus, the strong representation power of multilayer Transformers indeed comes at the price of robustness. This begs an important question: *is it possible to achieve both robustness and expressivity at the same time?*

Recently, there has been interest in the special class of Transformers, called Looped Transformers, which have empirically shown to have much more robust out-of-distribution generalization (Yang et al., 2023). Looped Transformers are essentially multilayer Transformers that share weights across layers. It is natural to ask if Looped Transformers exhibit similar representation-robustness tradeoff as multilayer Transformers. Perhaps surprisingly, we provide a negative answer to this question. In particular, we show that Looped Transformers can match the lower bound of $\log(\kappa)$ on the representative power, similar to multilayer Transformers. However, this does not come at the cost of robustness. In fact, Looped Transformers can achieve very good out-of-distribution generalization. In light of this discussion, we state the following main contributions of our paper.

- We consider the task-diverse in-context linear regression where the eigenvalues of the input covariance matrix has eigenvalues in the range $[1, \kappa]$; to handle this class of instances, we show a $\Omega(\sqrt{\kappa})$ lower bound on the number of required attention heads in the restricted attention case (Section 2.3), and a $\Omega(\log(\kappa))$ lower bound for the unconstrained case (Theorem 1). The $\Omega(\log(\kappa))$ lower bound matches the upper bound presented in Fu et al. (2023). In particular, our lower bounds hold for linear attention as well as for ReLU non-linearity.

- We show a matching upper bound on the constrained case in equation 5 by proving that the Transformer can implement multiple steps of the Chebyshev iteration algorithm (Theorem 2).

- We design an adaptive termination condition for multilayer Transformers based on the norm of the residuals; thereby, providing an flexible and efficient mechanism for early stopping with fewer layers (Theorem 3).

- We show that for certain training distributions on the covariance matrix, multilayer Transformers can overfit, in which case even if we are at a global minimizer with zero population loss and the test distribution has exponentially small deviation of $O(Ld^{-1}9^{-L})$ in Wasserstein distance from the training distribution, the test loss can be up to a constant error (Theorem 6).

- In contrast to the multilayer case, we show that under a mild condition that the training distribution puts some non-trivial mass on covariance matrices with large eigenvalues, the global minimizer of the training loss for loop Transformers extrapolate to most other distributions that have the same support (Theorem 7).

- Our theoretical insights regarding how depth make Transformers more task-diverse, and the robust out-of-distribution generalization of Looped Transformers compared to standard multilayer Transformers are validated through experiments in simple linear regression setting.

## 1.1 RELATED WORK

**Transformers in Task Diversity.** Raventós et al. (2024) observe a connection when the level of task diversity in the pretraining distribution is above a threshold, and the emergence of strong in-context learning abilities for linear regression, by switching from a Bayesian estimator to ridge regression. Garg et al. (2022); Akyürek et al. (2022); Von Oswald et al. (2023) observe that Transformers can extrapolate to new test distributions for in-context learning. Min et al. (2022) discover surprising latent task inference capabilities in language models, even when using random labels for in-context demonstrations. Kirsch et al. (2022) empirically show the benefits of task diversity for in-context learning in classification tasks. Chan et al. (2022) investigate which aspects of the pretraining distribution enhance in-context learning and find that a more bursty distributions proves to be more effective than distributions that are closer to uniform. The impact of task diversity has also been studied in the context of meta learning (Yin et al., 2019; Kumar et al., 2023; Tripuraneni et al., 2021) or transfer learning and fine-tuning (Raffel et al., 2020; Gao et al., 2020). In the current work, we use task diversity to study the expressivity of Transformers and the role of depth.

**Transformers and Looping as Computational Models.** Transformers are known to be effective computational models and Turing-complete (Pérez et al., 2019; 2021; Giannou et al., 2023). Garg et al. (2022); Akyürek et al. (2022); Von Oswald et al. (2023); Dai et al. (2022) argue that Transformers can implement optimization algorithms like gradient descent by internally forming an appropriate loss function. Allen-Zhu and Li (2023) observe that Transformers can generate sentences in Context-free Grammers (CFGs) by utilizing dynamic programming. Other work focuses on how training algorithms for Transformers can recover weights that implement iterative algorithms, such as gradient descent, for in-context learning, assuming simplified distributions for in-context instances Ahn et al. (2024b); Zhang et al. (2024); Mahankali et al. (2023). Notably, while most of these studies focus on the single-layer case, recent work by the authors in Gatmiry et al. (2024) provides a global convergence result for training deep Looped Transformers.

## 2 PRELIMINARIES

### 2.1 IN-CONTEXT LINEAR REGRESSION

In this works, we study the impact of depth, looping, and diversity on in-context learning for the supervised learning task of linear regression. The setting of linear regression for in-context learning has also been studied by Gatmiry et al. (2024); Ahn et al. (2024a); Von Oswald et al. (2023); Akyürek et al. (2022). We briefly describe the setup here.

**Linear regression.** Each supervised learning task in our setting is given by a linear regression instance $(X, w^*, y, x_q)$ with data matrix $X \in \mathbb{R}^{d \times n}$, labels $y$, regressor $w^*$, and query vector $x_q$. In particular, $X$ consists of the observed data inputs $\{x_i\}_{i=1}^n$ as its columns: $X = [x_1, \ldots, x_n]$. In this work, we assume that the instances are *realizable* i.e., $X^\top w^* = y$. Thus, $w^*$ can be obtained by the following pseudo-inverse relation:

$$w^* = (XX^\top)^{-1}Xy.$$

We define $\Sigma = XX^\top$ to be the input covariance since this will play an important role going forward.

## 2.2 TASK DIVERSITY & ROBUSTNESS

To handle task diversity, we need a model that not only makes accurate predictions for a fixed distribution, but is also robust to a class of distributions. Here, we focus on the notion of robustness with respect to various distributions on the covariance matrix $\Sigma$ of the linear regression instance. It is unreasonable to expect the model to behave well for all possible distributions, so we focus on the set of instances where the eigenvalues of the covariance matrix are in the range $(\alpha, \beta)$ for $0 < \alpha < \beta$.

**Definition 1.** *We define the set $S_{\alpha,\beta}$ of $(\alpha, \beta)$ "normal" linear regression instances as the set instances with a well-conditioned covariance matrix $X^\top X$:*

$$S_{\alpha,\beta} = \left\{ (X, w^*, y, x_q) : \alpha I \preccurlyeq XX^\top \preccurlyeq \beta I \right\}.$$

In order for the model to handle task diversity on the set of instances in $S_{\alpha,\beta}$, the loss needs to be small for any distribution $P_{\alpha,\beta}$ that is supported on covariances $\alpha I \preccurlyeq \Sigma \preccurlyeq \beta I$, which means the model has to be accurate for all the instances in $S_{\alpha,\beta}$. In this work, we are interested in understanding the ability of Transformers to handle task diversity from two angles: (1) expressiveness of the Transformer architecture is in representing task-diverse models (Section 3) and (2) robustness of these models in terms of out-of-distribution generalization. We first formally define the models used in this paper before exploring these questions.

## 2.3 TRANSFORMER MODEL

**Self-Attention layer.** Central to our paper is the single attention layer, which we define below. First we define the matrix $Z^{(0)}$ as a $(d+1) \times (n+1)$ matrix by combining the data matrix $X$, their labels $y$, and the query vector $x_q$ as follows: $Z^{(0)} = \begin{bmatrix} X & x_q \\ y^\top & 0 \end{bmatrix}$.

Given the key, query, and value matrices $W_k, W_q, W_v$, we follow (Ahn et al., 2024a; Von Oswald et al., 2023) and define the self-attention model $\text{Attn}^{lin}(Z; W_{k,q,v})$ with activation function $\sigma$ as:

$$\text{Attn}^{lin}(Z; W_{k,q,v}) \triangleq W_v Z M \sigma(Z^\top W_k^\top W_q Z),$$

$$M \triangleq \begin{bmatrix} I_{n \times n} & 0 \\ 0 & 0 \end{bmatrix} \in \mathbb{R}^{(n+1) \times (n+1)},$$

where the index $k \times r$ below a matrix determines its dimensions, and by $\sigma(\mathfrak{M})$ for matrix $\mathfrak{M}$ we mean entry-wise application of $\sigma$ on $\mathfrak{M}$. To simplify the notation for analysis, we combine the key and query matrices into $Q$, as $Q = W_k^\top W_q$. Hence, we have the following alternative parameterization:

$$\text{Attn}^{lin}(Z; Q, P) \triangleq PZM\sigma(Z^\top Q Z),$$

where now the learnable parameters are the matrices $P, Q$. We consider the cases where $\sigma(x) = Relu(x)$ or $\sigma(x) = x$ is linear. If $\sigma$ is linear, then the model becomes linear self-attention, which was first considered by Ahn et al. (2024a); Von Oswald et al. (2023); Schlag et al. (2021) to understand the behavior of in-context learning for linear regression. Note that even with linear attention, the output of the attention layer $\text{Attn}^{lin}(Z; Q, P)$ is a nonlinear map in $(P, Q)$ or $Z$, and hence challenging to analyze. Since we consider multilayer models, it turns into a low-degree polynomial in either $Z$ or in the set of weights $\{P^{(t)}, Q^{(t)}\}_{t=0}^{L-1}$; the limits of its expressivity is investigated in this work. Furthermore Ahn et al. (2024b) showed that studying linear attention can already provide signal about non-linear attention. Here, we study the questions of expressivity and robustness.

**Transformers.** Using $L$ attention heads, we define a Transformer block $\text{TF}(Z^{(0)}, \{P^{(t)}, Q^{(t)}\}_{t=0}^{L-1})$. In particular, following the notation in (Ahn et al., 2024a), we define

$$Z^{(t+1)} \triangleq Z^{(t)} - \tfrac{1}{n}\text{Attn}^{lin}\left(Z^{(t)}; Q^{(t)}, P^{(t)}\right). \tag{1}$$

which results in the recursive relation

$$Z^{(t+1)} = Z^{(t)} - \tfrac{1}{n}P^{(t)}Z^{(t)}M\sigma(Z^{(t)^\top}Q^{(t)}Z^{(t)}). \tag{2}$$

Then, the final output of the Transformer is the $((d+1),(n+1))$ entry of $Z^{(t)}$:

$$\text{TF}(Z^{(0)}, \{P^{(t)}, Q^{(t)}\}_{t=0}^{L-1}) = -Z^{(L)}{}_{(d+1),(n+1)}. \tag{3}$$

Note the minus sign in the final output, which is primarily for the ease of our exposition later on.

**Looped Transformer Model.** A Looped Transformer model is simply a multilayer Transformer with weight-sharing i.e., we define it as $\text{TF}(Z^{(0)}, P, Q) = -Z^{(L)}{}_{(d+1),(n+1)}$ where

$$Z^{(t+1)} = Z^{(t)} - \tfrac{1}{n} P Z^{(t)} M \sigma(Z^{(t)\top} Q Z^{(t)}). \tag{4}$$

**Restricted Attention.** Following Ahn et al. (2024a), in addition to our lower bound in Theorem 1 for the more general case when $\{P^{(t)}, Q^{(t)}\}_{t=0}^{L-1}$ can be arbitrary, we also study a relevant special case when the last row and column of $Q^{(t)}$ are zero and only the last row of $P^{(t)}$ is non-zero i.e.,

$$Q^{(t)} = \begin{bmatrix} A^{(t)} & 0 \\ 0 & 0 \end{bmatrix}, \; P^{(t)} = \begin{bmatrix} 0_{d\times d} & 0 \\ u^{(t)\top} & 1 \end{bmatrix}. \tag{5}$$

This special case is important as it can implement preconditioned gradient descent, while it keeps the feature matrix $X$ the same going from $Z^{(t-1)}$ to $Z^{(t)}$. In this case, the left upper $d \times n$ block of the second term in equation 2 is always zero. This is because we assume the $d \times d$ zero block in $P^{(t)}$. We denote the output of the Transformer in this case by $\text{TF}(Z^{(0)}, \{A^{(t)}, u^{(t)}\}_{t=0}^{L-1})$. We denote the first $n$ entries of the last row of $Z^{(t)}$ by vector $y^{(t)\top}$, and the last entry by $y_q^{(t)}$, i.e. $y_q^{(t)} = Z^{(t)}{}_{(d+1)\times(n+1)}$. Unrolling equation 5, we obtain the following recursions for $y^{(t)}$ and $y_q^{(t)}$:

$$y^{(t+1)\top} = y^{(t)\top} - \frac{1}{n}(y^{(t)\top} + u^{(t)\top}X)\sigma\left(X^\top A^{(t)} X\right),$$

$$y_q^{(t+1)} = y_q^{(t)} - \frac{1}{n}(y^{(t)\top} + u^{(t)\top}X)\sigma\left(X^\top A^{(t)} x_q\right).$$

Note that the final output of the Transformer, $\text{TF}(Z^{(0)}, \{A^{(t)}, u^{(t)}\}_{t=0}^{L-1})$, is equal to $y_q^{(L)}$.

# 3 ON THE ROLE OF DEPTH IN MULTILAYER TRANSFORMERS FOR TASK DIVERSITY

In this section, we examine the power and limits of multilayer Transformers. We first present lower bounds for multilayer Transformers in both the general case and the restricted attention case (Section 2) and, then provide matching upper bounds. While this makes a compelling case for the expressivity power of multilayer Transformers, in Section 4.2.1, we show that multilayer Transformers can suffer from overfitting and perform very poorly under distribution shifts.

## 3.1 LOWER BOUND ON THE POWER OF MULTILAYER TRANSFORMERS

In this section, we provide lower bound for multilayer Transformers in solving in-context learning.

**Lemma 1** (Effect of scaling on accuracy - restricted attention)**.** *For an $L$-layer Transformer with architecture defined in equation 2 and 5, with arbitrary weights and positive homogeneous ReLU activation $\sigma$ or simply using linear attention ($\sigma(x) = x$), consider an arbitrary instance of a realizable linear regression $(X, w^*, y, x_q)$;*

1. *There exists a scaling $1 \le \gamma \le 36L^2$ such that the Transformer's response for the scaled instance $(\gamma X, w^*, \gamma y, x_q)$ will be inaccurate by constant:*

$$\frac{|TF(Z^{(0)}, \{A^{(t)}, u^{(t)}\}_{t=0}^{L-1}) - w^{*\top}x_q|}{|w^{*\top}x_q|} \ge \frac{1}{4}, \tag{6}$$

*where $Z^{(0)} = \begin{bmatrix} \gamma X & x_q \\ \gamma y^\top & 0 \end{bmatrix}$ is the Transformer input corresponding to the instance $(\gamma X, w, \gamma y, x_q)$. Even more, there exists an interval $[a, b]$ inside $[1, 36L^2]$ with length at least constant (independent of $L$) such that for all $\gamma \in (a, b)$ we have equation 6.*

2. *Furthermore, if we do not restrict the weight matrices $\{P^{(t)}, Q^{(t)}\}_{t=0}^{L-1}$ to have the form in equation 5, then there is a scaling $1 \leq \gamma \leq 2^L$ such that*

$$\frac{|TF(\gamma Z^{(0)}, \{P^{(t)}, Q^{(t)}\}_{t=0}^{L-1}) - w^{*\top} x_q|}{|w^{*\top} x_q|} \geq \frac{1}{4}.$$

**Theorem 1** (Lower bound on the representation power of Transformers). *For the Transformer architecture with ReLU or linear activation defined in equation 3, under the weight restriction equation 5, consider the set $\mathcal{S}_{\alpha,\beta}$ of $(\alpha,\beta)$-normal realizable instances of linear regression $(X, w, y, x_q)$ where $\alpha I \preccurlyeq X^\top X \preccurlyeq \beta I$. Then, given $L \leq \sqrt{\beta/\alpha}$ and for any choice of $\{A^{(t)}, u^{(t)}\}_{t=1}^L$ and query vector $x_q$, there exists a normal instance $(X, w^*, y, x_q) \in \mathcal{S}_{\alpha,\beta}$ with that query vector, such that*

$$\frac{|TF(Z^{(0)}, \{A^{(t)}, u^{(t)}\}_{t=0}^{L-1}) - w^{*\top} x_q|}{|w^{*\top} x_q|} \geq \frac{1}{2}. \tag{7}$$

*Furthermore, beyond the restricted attention ( equation 5), given $L = O\left(\log\left(\beta/\alpha\right)\right)$ for small enough constant, we incur at least constant error for some instance $(X, w^*, y, x_q) \in \mathcal{S}_{\alpha,\beta}$.*

Theorem 1 asserts that a minimum depth is required for multilayer Transformer to be able to solve all instances of linear regression whose covariance matrix is in the range $[\alpha, \beta]$. In particular, this lower bound depends on $\frac{\beta}{\alpha}$, which increases with the diversity of the instances. For the restricted attention case, the lower bound is $\Omega(\sqrt{\beta/\alpha})$ whereas for the general case the lower bound is $\Omega(\log(\beta/\alpha))$.

**Remark 3.1.** *(Fu et al., 2023) show that, given full freedom in the weights of Transformers, one can implement one step of the iterative Newton method with an attention head. Hence, multilayer Transformers with depth at least $\Omega(\ln\left(\beta/\alpha\right))$ are able to accurately solve all instances in $S_{\alpha,\beta}$. Our result essentially shows a matching lower bound for this case in Theorem 1, for achieving any constant error bound that can be arbitrarily small. Interestingly, their construction uses the same set of weights for all of the attention layers, therefore, their network is indeed a looped Transformer. This shows that looped models can also match the lower bound in terms of numbers of layers required. We will revisit this in Section 4.1.*

## 3.2 MATCHING UPPER BOUND FOR MULTILAYER TRANSFORMER

We show the existence of weights for a linear Transformer that can solve a linear regression instance with bounded condition number using restricted attention defined in equation 5.

**Theorem 2.** *For $\sigma(x) = x$, there exists a set of weights $\{A^{(t)}, u^{(t)}\}_{t=0}^{L-1}$ for linear Transformer in the restricted case defined in equation 5 with depth at most $L = O(\ln(1/\epsilon)\sqrt{\beta/\alpha})$ such that for every linear regression instance $(X, w^*, y, x_q) \in \mathcal{S}_{\alpha,\beta}$ we have*

$$|TF(Z^{(0)}, \{A^{(t)}, u^{(t)}\}_{t=0}^{L-1}) - w^{*\top} x_q| \leq \epsilon \|w^*\| \|x_q\|.$$

Notably, the dependency $\sqrt{\beta/\alpha}$ in Theorem 2 matches the lower bound that we show for the same restricted Transformer model in Theorem 1. The square root dependency comes from the fact that one can implement an iterative algorithm for solving a linear regression instance using the properties of Chebyshev polynomials. The proof of Theorem 2 can be found in Section C.3. The $\sqrt{\beta/\alpha}$ dependence is indeed reminisent of the accelerated rate dependency on the condition number in smooth ans strongly convex optimization.

## 3.3 ADAPTIVE DEPTH

In this section, we investigate adaptive strategies that can be used for terminating a multilayer Transformer earlier, before passing the input through all the $L$ layers. Surprisingly, our result below, shows convergence of the output of the Transformer if we adaptively terminate based on $\|y^{(\ell)}\|$.

**Theorem 3** (Termination guarantee). *For a linear Transformer architecture as defined in equation 2 with $\sigma(x) = x$, given $u^{(i)} = 0$, suppose we wait until $\|y^{(\ell)}\| \leq \frac{\epsilon}{\|x_q\|_{XX^\top}}$. Then, the output of the Transformer at layer $\ell$ is close to the true label:*

$$|y_q - TF(Z^{(0)}, \{A^{(t)}, u^{(t)}\}_{t=0}^{\ell-1})| \leq \epsilon \|x_q\|_{XX^\top}.$$

This result further highlights the computational benefits by terminating in an adaptive manner.

## 4 ON THE POWER & ROBUSTNESS OF LOOPED TRANSFORMERS VERSUS MULTI-LAYER TRANSFORMERS

In the previous section, we studied expressivity of multilayer Transformers for task-diverse linear regression. In Section 4.1, we discuss how even though Looped Transformers are more restricted compared to multilayer Transformers, their expressive power is just as good for in-context linear regression. While expressivity is important for task-diversity of our model, we also want the model to be robust to distribution shifts from the training distribution. In Section 4.2, we show that multilayer Transformers are not very robust to even very tiny perturbations to the training distribution. In contrast, Looped Transformers are provably more robust. Finally, motivated by the fact that early stopping is a desirable property for neural nets, in Section 4.3 we study the monotonicity behavior of multilayer Transformers with respect to depth and interestingly find that they cannot behave monotonic with respect to all covariance matrices unless the weights of different layers are equal.

### 4.1 EXPRESSIVITY OF LOOPED TRANSFORMERS

Looped Transformers are a subclass of multilayer Transformers that uses weight-sharing, and thus, cannot exceed multilayer Transformers in representation power. Despite this restriction, we observe that their ability in handling task-diversity almost remains unchanged. In particular, according to the construction in (Fu et al., 2023), there exists a Looped transformer in addition to constant number of attention layers, that can achieve small error for all instances in $S_{\alpha,\beta}$ after $\ln(\beta/\alpha)$ number of loops, matching the lower bound in Theorem 1.

**Theorem 4** (Restatement of Theorem 5.1 in Fu et al. (2023)). *There exists a Looped Transformer with additional eight attention layers that implements the Newton algorithm i.e., for instance $(X, w^*, y, x_q)$, looping $L$ times results in output $\hat{x}_q^\top w_L^{(Newton)}$ where $w_L^{(Newton)} \triangleq M_L X y$ where $M_j$ is updated as $M_j = 2M_{j-1}\Sigma M_{j-1}, j \in [1, L], M_0 = \alpha\Sigma$, where $\Sigma = XX^\top$.*

Furthermore, in the restricted attention case as defined in equation 5, with the implementation of gradient descent by Von Oswald et al. (2023) which also uses weight sharing, it is easy to check that Looped Transformers can still achieve small error on all instances in $S_{\alpha,\beta}$ with depth $O(\beta/\alpha)$ (see Section B); in this case, we see a gap with the $\Omega(\sqrt{\beta/\alpha})$ lower bound in Theorem 1.

**Theorem 5** (Follows from Proposition 1 in Von Oswald et al. (2023)). *There exists a Looped Transformer architecture in the restricted attention case which for a linear regression instance $S = (X, w^*, y, x_q) \in \mathcal{S}_{\alpha,\beta}$ can achieve accuracy $|TF(Z^{(0)}, \{A, 0\}_{t=0}^{L-1}) - y| \le \epsilon$ after $O(\ln(1/\epsilon)\beta/\alpha)$ number of loops.*

### 4.2 OUT-OF-DISTRIBUTION GENERALIZATION

So far we examined the crucial role of depth for task-diversity settings. However, it is unclear if the weights that minimize the population loss are robust to distributions shifts. To this end, we study the task diversity of transformers through the lens of out-of-distribution generalization. We first study the limitations of multi-layer Transformers with respect to out-of-distribution generalization. Recall that each instance of the in-context learning problem is a linear regression supervised-learning task, denoted by $(X, y, w^*, x_q)$, so a population distribution over such instances corresponds to a distribution over $X, w^*, x_q$. To enable this study, we consider the population loss with respect to various distributions on the covariance matrix in the restricted attention case defined in Section 2.3. In particular, we focus on distribution shifts on the covariance matrix of the linear regression instances, so we set $u^{(t)} = 0$ and define the loss function with respect to a distribution $P$ on the covariance matrix $\Sigma = XX^\top$. For simplicity, we further assume that the population distribution over $x_q$ and $w^*$ are according to $N(0, \Sigma)$ and $N(0, \Sigma^{-1})$, respectively. This choice of distribution follows (Gatmiry et al., 2024). Note that while this breaks the independence assumption between $x_i$'s and $x_q$, when the number of samples $n$ grows large, given that the sample covariance matrix

converges to the population covariance $\Sigma^*$, then this assumption reduces to the i.i.d case where $x_i, x_q \sim N(0, \Sigma)$. Therefore, we define the population loss as

$$\mathcal{L}^P(\{A^{(t)}\}_{t=0}^{L-1}) \triangleq \mathbb{E}_{\Sigma \sim P, w^* \sim N(0, \Sigma^{-1}), x_q \sim N(0, \Sigma)} \left( \text{TF}(Z^{(0)}, \{A^{(t)}, 0\}_{t=0}^{L-1}) - y \right)^2. \quad (8)$$

In particular, for the special case of the looped model (where $A^{(t)} = A$ for $t \in [L-1]$), the loss is:

$$\mathcal{L}^P(A) \triangleq \mathbb{E}_{\Sigma \sim P, w^* \sim N(0, \Sigma^{-1}), x_q \sim N(0, \Sigma)} \left( \text{TF}(Z^{(0)}, \{A, 0\}_{t=0}^{L-1}) - y \right)^2.$$

When the distribution $P$ is a point mass on covariance $\Sigma$, we denote the loss by $\mathcal{L}^\Sigma(\{A^{(t)}\}_{t=0}^{L-1})$ and $\mathcal{L}^\Sigma(A)$. We denote a distribution on the covariance matrices that is only supported on $\{\alpha I \preccurlyeq \Sigma \preccurlyeq \beta I\}$ by $P_{\alpha, \beta}$, with sub-indices $\alpha, \beta$ showing the interval of the eigenvalues for the support of the covariances. The key message that we deliver in this section is that loop Transformers are more robust for out-of-distribution generalization compared to multilayer Transformers. To elucidate the advantage of looped over multilayer Transformers, we need the following definition (Section 4.2.2).

**Definition 2.** *We say the distribution $P_{\alpha, \beta}$ supported on the covariance matrices of linear regression instances in $S_{\alpha, \beta}$ (or we abbreviate by saying supported on $S_{\alpha, \beta}$) is $(\epsilon, \delta)$ right-spread-out if for every fixed unit vector $v$, $P_{\alpha, \beta}(\|X^\top v\|^2 \geq (1-\delta)\beta) \geq \epsilon$.*

The right-spread-out property restricts the distribution on the data covariance $XX^\top$ so that for every direction $v \in \mathbb{R}^d$, it puts a minimum amount of mass on matrices whose eigenvectors with large eigenvalues, that are close to the right end point of the interval $(\alpha, \beta)$, are close to $v$.

### 4.2.1 Weakness of Multilayer Transformers for Out-of-Distribution Generalization

From Section 3.2, recall that there exists a multilayer Transformer of depth $O(\ln(1/\epsilon)\sqrt{\beta/\alpha})$ that can solve any linear regression with covariance $\Sigma$ such that $\alpha I \preccurlyeq \Sigma \preccurlyeq \beta I$. However, it is unclear if minimizing the training loss recovers such a network. Here, we show that this is not the case.

**Theorem 6** (Multilayer Transformers blow up out of distribution). *Given $\frac{\beta}{\alpha} \geq 10$ and for any $\epsilon > 0$, there exists a $(\frac{1}{L}, 0)$ right-spread-out distribution $P_{\alpha, \beta}$ such that there exists a global minimizer $\{A^{(t)^*}\}_{t=0}^{L-1}$ of the loss $\mathcal{L}^{P_{\alpha,\beta}}(\{A^{(t)}\}_{t=0}^{L-1}, 0)$ for linear Transformer defined in equation 2, so that for any other distribution $\tilde{P}_{\alpha, \beta}$ on $\Sigma$ which has at least $\epsilon$ mass supported on $8\alpha I \preccurlyeq \Sigma \preccurlyeq (1-\delta')\beta I$, the out of distribution loss $\mathcal{L}^{(\tilde{P}_{\alpha,\beta})}$ exponentially blows up at $\{A^{(t)^*}\}_{t=0}^{L-1}$ with the depth:*

$$\mathcal{L}^{(\tilde{P}_{\alpha,\beta})}(\{A^{(t)^*}\}_{t=0}^{L-1}, 0) \geq \epsilon \delta' d 9^{L-1}.$$

We further suspect that there is only one global minimizer of the loss in Theorem 6. In particular, Theorem 6 shows that multilayer Transformers are essentially prone to overfitting, so that with a slight deviation of the test distribution from the train distribution (even $O(\frac{L}{d9^{L-1}})$) in Wasserstein distance, the test loss will incur constant error for the global minimizer of the train loss.

### 4.2.2 Out-of-distribution generalization of Looped Transformers

Our result in Theorem 6 states that multi-layer Transformers can behave poorly for out-of-distribution generalization. Here, we show that Looped Transformers can indeed circumvent this issue by restricting the model with weight sharing when the training distribution is right-spread-out.

**Theorem 7** (Looped Transformer is robust out-of-distribution). *Given an $(\epsilon, \delta)$ right-spread-out distribution $P_{\alpha, \beta}$ on $S_{\alpha, \beta}$, let $A^*$ be the global minimizer of $\mathcal{L}^{(P_{\alpha,\beta})}(A, 0)$ for the linear Looped Transformer defined in equation 4 with $\sigma(x) = x$ in the region $\alpha I \preccurlyeq A^{-1} \preccurlyeq \beta I$. Then, given any $\epsilon' \leq \epsilon$ and any other distribution $\tilde{P}_{\alpha, (1-\delta')\beta}$ that is supported on $(\alpha, (1-\delta')\beta)$, for*

$$\delta' = \delta + \frac{\ln(d/\epsilon')}{L} \text{ such that } \delta' \leq 1 - \frac{\alpha}{\beta}, \text{ we have } \mathcal{L}^{(\tilde{P}_{\alpha,(1-\delta')\beta})}(A^*, 0) \leq \epsilon' \wedge \left(1 - \frac{\alpha}{\beta}\right)^{2L}.$$

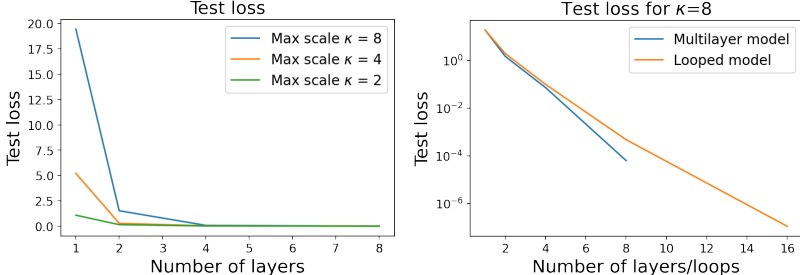

Figure 1: We evaluate the test loss of looped models and multilayer model as the function of loops and depth respectively. In (a), we plot the test loss as a function of number of layers for three different covariance ranges. As predicted by theory, the larger the range ($\kappa$), the more layers are needed to get a small loss. In (b) we observe that the number of loops required to solve the problem is very close to the number of layers required for multilayer model.

Theorem 7 demonstrates that if the training distribution is sufficiently spread out, the global minimizer in a looped model remains robust to a wide range of distributional shifts in the test loss. Specifically, the test distribution can be any arbitrary distribution on the covariance matrix, as long as its eigenvalues lie within the slightly narrowed interval of $(\alpha, (1 - \delta')\beta)$, where $\delta' = \delta + \gamma \ln(d)$ approaches $\delta$, the spread-out parameter of the training distribution, as the number of loops increases. This contrasts with the behavior of multilayer Transformers, which, as shown in Theorem 6, can overfit when trained on right-spread-out distributions.

### 4.3 NON-MONOTONICITY OF THE LOSS IN MULTILAYER TRANSFORMER

In Section 3.3, we discussed how one can adaptively pick a depth based on problem difficulty. A related idea is that of early-exiting (Teerapittayanon et al., 2016), where the goal is to exit the model at an earlier layer for easier example. This naturally provides inference efficiency. An important consideration in the early exit literature is monotonicity with respect to layers, i.e., the model's error decreases as the layer index increases. This has been studied in detail (Baldock et al., 2021; Laitenberger et al.) and there is also work on enforcing such monotonicity (Schuster et al., 2022; Jazbec et al., 2024). In the linear regression incontext learning setting considered in this paper, we make an intriguing observation: if a multilayer model has monotonically decreasing error with depth for a diverse set of distributions, then it must be a looped model. This suggests that looped models are naturally suited for early-exiting strategies. We believe this phenomenon deserves further exploration in future work. The following theorem formalizes this idea.

**Theorem 8** (Monotonic behavior w.r.t depth $\rightarrow$ equal weights). *For multi-layer Transformer $TF(\{A^{(t)}, 0\}_{t=0}^{L-1}, \{A^{(t)}, u^{(t)}\}_{t=0}^{L-1})$ if the average value of the loss is monotonic for $0 \leq t \leq L-1$ for every covariance matrix $\Sigma$, then we have $A^{(1)} = \cdots = A^{(L)}$. Moreover, for the loop model and for every covariance $\Sigma$, there exists an $L_0$ which depends on $\Sigma$, such that for all $L \geq L_0$, the behavior of the loss becomes monotonic with respect to $L$.*

Proof of Theorem 8 can be found in Section F. At a high level, Theorem 8 states that in order to have a multilayer Transformer in the restricted attention case (Section 2) for which the population loss behaves monotonically for an arbitrary choice of distribution on the covariance matrix, then the only possibility is for the Transformer to be looped. Furthermore, Theorem 8 states that this monotonicity property indeed holds for the looped model for large enough depth. The proof of Theorem 8 uses the fact that for any covariance distribution, the loss function can be related to the spectrum of the weight matrices $A^{(i)}$'s, and that having unequal $A^{(i)}$'s, one can construct an adversarial distribution on the covariance for which the loss does not behave monotonically with respect to the depth.

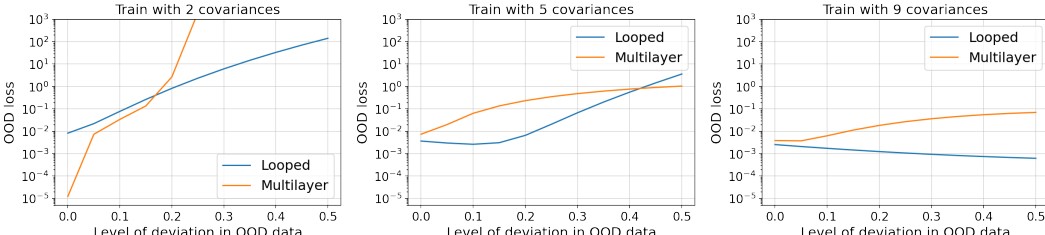

Figure 2: We evaluate the robustness of looped models and multilayer model as the level of deviation of the test distribution from the train distribution increases. The three plots differ in the number of distinct covariances used in the train distribution. The test distribution is supported on a window of size $w$ around the train covariances, and $w$ is varied on the x-axis. We see that in all settings the looped model is more robust than the multilayer model, especially on the left plot where there is least diversity in the training distribution, as predicted by the theory.

## 5 EXPERIMENTS

In this section, we run experiments on the in-context learning linear regression problem to validate the theoretical results. In particular, we would like to demonstrate: (1) role of depth (in both multilayer and Looped Transformers) with increasing task diversity and (2) robustness of looped Transformer for out-of-domain generalization. We use the codebase and experimental setup from (Ahn et al., 2024a) for all our linear regression experiments. In particular, we work with $d = 10$ dimensional inputs and each input instance consists of $n = 20$ pairs of $(x, y)$ in the context. We train with $L$ attention layer models for multilayer training and 1 layer attention model looped $L$ times, as described in Section 2.3. For simplicity we use the restricted linear attention setting. In all experiments, we follow the same data distribution as described in Section 4.2. For all experiments, the covariances are sampled from training and test distributions as follows: train distribution is $\Sigma = sI_d$, $s \sim \mathcal{D}_{\text{train}}$ and test distribution: $\Sigma = sI_d$, $s \sim \mathcal{D}_{\text{test}}$. In each experimental setting we will specify $\mathcal{D}_{\text{train}}$ and $\mathcal{D}_{\text{test}}$.

**Role of Depth.** We demonstrate the importance of depth in presence of task diversity. Consider the following train and test distributions $\mathcal{D}_{\text{train}} = \mathcal{D}_{\text{test}} = \text{Unif}([1, \kappa])$. Task diversity is controlled by varying $\kappa \in [2, 4, 8]$, with higher value corresponding to more diverse tasks. In Figure 1, we see that that more diverse tasks (larger $\kappa$) require more layers (or loops) in order to achieve low loss. This aligns with the theoretical results from Section 3.1. Furthermore, we find that looped model with $L$ loops can be very competitive to multilayer model with $L$ layers even in the most diverse setting.

**Out-of-distribution generalization.** For this set of experiments, we train models using a distribution comprising of $k$ covariances. The precise form the train and test distributions are:

- Train distribution: $\mathcal{D}_{\text{train}} = \text{Unif}(\{s_1, \ldots, s_k\})$ where $S_k = \{s_1, \ldots, s_k\}$ is selected to be $k$ values uniformly spread out in the range $[1, 8]$.

- Test distribution: $\mathcal{D}_{\text{train}} = \text{Unif}(\cup_{i=1}^{k}[s_i - w, s_i + w])$ for some deviation $w \in \mathcal{R}$.

Note that the train distribution is a mixture of $k$ tasks, whereas the test distribution has a slight deviation using a window size of $w$ around the training covariances. For evaluations, $w$ is varied from 0, corresponding to in-distribution evaluation, to a max value of 0.5. In Figure 2, we plot the robustness of multilayer and looped model for different training distributions corresponding to $k = 2, 5$ and 9. Consistent with our theory in Section 4.2, we observe that looped Transformers have much better out-of-distribution generalization compared to multilayer Transformers. This also aligns with the empirical results from Yang et al. (2023) on various incontext learning problems. Furthermore, as the number of train covariances increase, the looped model becomes perfectly robust.

## 6 CONCLUSION

In this paper, we study a more realistic in-context setting than prior works — task-diverse linear regression, which we believe is more reflective of the incontext abilities of Transformer based foundation models. For this setting, we provide lower bounds on the depth and show Transformers match these bounds; thereby, providing a comprehensive picture about number of layers required to solve the problem. While multilayer Transformers demonstrate this powerful representation power, we show they have weak out-of-distribution generalization, which gets exponentially worse with respect to depth. Surprisingly, we show the Looped Transformers exhibit similar representative power with much better robustness. We validate our theoretical findings through experiments on linear regression setting. Finally, we touch upon aspects like adaptive depth and monotonicity of loss in layers, and we believe that these are very interesting future directions to pursue to understand the power of looped models. While the incontext linear regression setting is quite simplified, it already provides a lot of interesting insights in the role of depth and task diversity. Extending this study to other settings, like reasoning, is a very interesting future direction.

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

## A  RELATED WORK

**Transformers in Task Diversity.**  Raventós et al. (2024) empirically study the relationship between task diversity in the pretraining distribution and the emergence of in-context learning abilities for linear regression. They observe an intriguing phase transition, where as task diversity increases, the learned model shifts from acting as a Bayesian estimator to implementing ridge regression. Garg et al. (2022) investigate in-context learning across various tasks, where both the inputs and the link function for the in-context instances are sampled from an underlying distribution.  For in-context linear regression, Garg et al. (2022); Akyürek et al. (2022); Von Oswald et al. (2023) further observe that transformers are capable of extrapolating to new test distributions, both in terms of the inputs and the regressor. Min et al. (2022) discover surprising latent task inference capabilities in language models; even when using random labels for in-context demonstrations (i.e., in the test distribution), the model can still produce accurate predictions, provided the input distribution in the test instances is similar to that of the pretraining data.  The impact of task diversity has also been studied in the context of meta learning Yin et al. (2019); Kumar et al. (2023); Tripuraneni et al. (2021). Kirsch et al. (2022) empirically demonstrate the benefits of task diversity for in-context learning in classification tasks. Chan et al. (2022) investigate which aspects of the pretraining distribution enhance in-context learning and find that a more bursty distribution, with potentially a large number of rare clusters, proves to be more effective than distributions that are closer to uniform. Raffel et al. (2020); Gao et al. (2020) explore the role of task diversity in the pretraining distribution for improvements in the context of transfer learning and fine-tuning on downstream tasks.

**Transformers as Computational Models.**  Pérez et al. (2021) Transformers are known to be powerful computational models, Pérez et al. (2019) demonstrating their Turing-completeness. Giannou et al. (2023) propose that Looped Transformers can function as programmable computers without scaling the size of its architecture with the runtime logic such as program loops. Akyürek et al. (2022); Von Oswald et al. (2023); Dai et al. (2022) argue that transformers are expressive enough to implement optimization algorithms like gradient descent by internally forming an appropriate loss function. Garg et al. (2022) empirically show that transformers can perform in-context learning across various learning tasks, while Allen-Zhu and Li (2023) observe that transformers can learn and generate sentences in Context-free Grammers (CFGs) by utilizing dynamic programming. Other work focuses on how training algorithms for Transformers can recover weights that implement iterative algorithms, such as gradient descent, for in-context learning, assuming simplified distributions for in-context instances Ahn et al. (2024b); Zhang et al. (2024); Mahankali et al. (2023). Notably, while most of these studies focus on the single-layer case, recent work by the authors in Gatmiry et al. (2024) provides a global convergence result for training deep Looped Transformers.

## B    REMAINING PROOFS

*Proof of Theorem 7.* First, note that the global minimizer $A^*$ of $\mathcal{L}$ and $A = \beta^{-1}I$ we should have $\mathcal{L}^{(P_{\alpha,\beta})}(A^*, 0) \leq \mathcal{L}^{(P_{\alpha,\beta})}(A, 0) \leq 1$. To see this, note that from the definition of optimality of $A^*$, we have $\mathcal{L}^{(P_{\alpha,\beta})}(A^*, 0) \leq \mathcal{L}^{(P_{\alpha,\beta})}(A, 0)$, and from the formula in Lemma 4:

$$\mathcal{L}(A, 0) = \mathbb{E}_{\Sigma \sim P_{\alpha,\beta}} \text{trace}\Big((I - \Sigma^{1/2} A \Sigma^{1/2})^{2L}\Big)$$

$$= \mathbb{E}_{\Sigma \sim P_{\alpha,\beta}} \text{trace}\big((I - \beta^{-1}\Sigma)^{2L}\big).$$

But since $\Sigma \preccurlyeq \beta I$, we get $I - \beta^{-1}\Sigma \preccurlyeq I$, hence $(I - \beta^{-1}\Sigma) \preccurlyeq I$, which implies

$$\mathcal{L}(A^*) \leq d. \tag{9}$$

Now we show that if given this upper bound on the optimal loss, we should necessarily have

$$A^{*-1} \succcurlyeq \frac{(1-\delta)\beta}{2}(1 - \frac{\ln(1/\epsilon)}{L})I.$$

Suppose this is not the case; Then, there is unit direction $v$ such that

$$v^\top A^{*-1} v < \frac{(1-\delta)\beta}{2}(1 - \frac{\ln(1/\epsilon)}{L}). \tag{10}$$

Now from the $(\epsilon, \delta)$-right-spread-out assumption, with probability at least $\epsilon$ we have

$$v^\top \Sigma v = v^\top X X^\top v \geq (1-\delta)\beta. \tag{11}$$

Now we claim that under this event we should necessarily have

$$\left\| I - \Sigma^{1/2} A^* \Sigma^{1/2} \right\| > 1 + \frac{\ln(d/\epsilon)}{L}.$$

Suppose this is not the case. Then

$$\Sigma^{1/2} A^* \Sigma^{1/2} \preccurlyeq (2 + \frac{\ln(d/\epsilon)}{L})I.$$

But this implies

$$(2 + \frac{\ln(d/\epsilon)}{L})A^{*-1} \succcurlyeq \Sigma,$$

which gives

$$(2 + \frac{\ln(d/\epsilon)}{L})v^\top A^{*-1} v \succcurlyeq v^\top \Sigma v.$$

But this contradicts equation 10 and equation 11 as

$$(2 + \frac{\ln(d/\epsilon)}{L})(1 - \frac{\ln(d/\epsilon)}{L})(1-\delta)\beta = (1 - \frac{\ln(d/\epsilon)}{L} - \left(\frac{\ln(d/\epsilon)}{L}\right)^2)(1-\delta)\beta < (1-\delta)\beta.$$

Hence, we should have

$$\left\| I - \Sigma^{1/2} A^* \Sigma^{1/2} \right\| > 1 + \frac{\ln(d/\epsilon)}{L}.$$

But this means that under this event, which happens with probability at least $\epsilon$ under $P_{\alpha,\beta}$, we should also have

$$\left(I - \Sigma^{1/2} A^* \Sigma^{1/2}\right)^{2L} \succcurlyeq \left(1 + \frac{\ln(d/\epsilon)}{L}\right)^L > \frac{d}{\epsilon}.$$

This implies

$$\mathcal{L}(A^*, 0) = \mathbb{E}_{\Sigma \sim P_{\alpha,\beta}} \text{trace}\Big((I - \Sigma^{1/2} A^* \Sigma^{1/2})^{2L}\Big)$$

$$\geq \Pr(v^\top \Sigma v \geq (1-\delta)\beta) \times \frac{d}{\epsilon} > d,$$

which contradicts equation 9. Therefore, we should have

$$A^{*-1} \succcurlyeq \frac{(1-\delta)\beta}{2}(1 - \frac{\ln(d/\epsilon)}{L})I. \tag{12}$$

Now note that for every $\Sigma$ sampled from $P_{\alpha,(1-\delta')\beta}$, according to definition we have

$$\Sigma \preccurlyeq (1-\delta')\beta I,$$

which combined with equation 12 implies

$$\Sigma^{1/2}A^*\Sigma^{1/2} \le \frac{(1-\delta')\beta}{\frac{(1-\delta)\beta}{2}(1 - \frac{\ln(d/\epsilon')}{L})} =$$

$$\le 2(1 - \frac{\ln(d/\epsilon)}{L}).$$

The last inequality is because

$$1 - \delta' = 1 - \delta - 2\frac{\ln(d/\epsilon')}{L} \le 1 - \delta - 2(1-\delta)\frac{\ln(d/\epsilon')}{L} + (1-\delta)\left(\frac{\ln(d/\epsilon')}{L}\right)^2 = (1-\delta)\left(1 - \frac{\ln(d/\epsilon')}{L}\right)^2.$$

On the other hand, note that for any $\Sigma$ in the support of $P_{\alpha,(1-\delta')\beta}$, we have

$$\alpha I \preccurlyeq \Sigma.$$

Therefore,

$$\left(\frac{\alpha}{\beta} - 1\right)I \preccurlyeq \Sigma^{1/2}A^*\Sigma^{1/2} - I \preccurlyeq \left(1 - \frac{2\ln(d/\epsilon')}{L}\right)I,$$

which implies

$$\left\|\left(I - \Sigma^{1/2}A^*\Sigma^{1/2}\right)^{2L}\right\| \le \left(1 - \left(\frac{2\ln(d/\epsilon')}{L} \wedge \frac{\alpha}{\beta}\right)\right)^{2L} \le \frac{\epsilon'}{d} \vee \left(1 - \frac{\alpha}{\beta}\right)^L.$$

Therefore

$$\mathcal{L}^{(\tilde{P}_{\alpha,(1-\delta')\beta})}(A^*, 0) = \mathbb{E}_{\Sigma \sim P_{\alpha,\beta}}\text{trace}\left((I - \Sigma^{1/2}A^*\Sigma^{1/2})^{2L}\right) \le \epsilon' \vee d\left(1 - \frac{\alpha}{\beta}\right)^L.$$

$\square$

*Proof of Theorem 5.* As described in Appendix A.1 in Von Oswald et al. (2023), we can construct one iteration of gradient descent for the linear regression quadratic loss in the restricted attention case, which takes the following form:

$$y^{(t+1)^\top} = y^{(t)^\top} - \eta y^{(t)^\top}X^\top X,$$

$$y_q^{(t+1)} = y_q^{(t)} - \eta y^{(t)^\top}X^\top x_q.$$

Then, taking the step size $\eta = \frac{1}{2\beta}$, for the linear regression error we get

$$|\text{TF}(Z^{(0)}, \{A, 0\}_{t=0}^{L-1}) - y| = w^{*\top}\left(I - \eta(XX^\top)\right)^L x_q \le \|w^*\|\|x_q\|(1 - \frac{\alpha}{2\beta})^L,$$

which completes the proof. $\square$

## C  LOWER BOUND

### C.1  A FORMULA FOR THE LOSS IN THE MULTILAYER CASE

**Lemma 2.** *[Output with non-linearity] We have the following relations*

$$y^{(t+1)^\top} = y^\top \prod_{i=0}^{t}\left(I - \frac{1}{n}\sigma\left(X^\top A^{(i)}X\right)\right),$$

$$y_q^{(t+1)} = \sum_{j=1}^{t-1} y^\top\left(I - \prod_{i=0}^{j-1}\left(I - \frac{1}{n}\sigma\left(X^\top A^{(i)}X\right)\right)\right)\sigma\left(X^\top A^{(j)}x_q\right) + \prod_{i=0}^{t-1}\left(I - \frac{1}{n}\sigma\left(X^\top A^{(i)}X\right)\right)\sigma\left(X^\top A^{(t)}x_q\right).$$

*Proof.* Assuming the formula for $y^{(t)}$ and from the update equation for $y^{(t+1)}$,

$$y^{(t+1)^\top} = y^{(t)^\top} - \frac{1}{n} y^{(t)^\top} \sigma \left( X^\top A^{(t)} X \right)$$

$$= y^{(t)\top} \left( I - \frac{1}{n} \sigma \left( X^\top A^{(t)} X \right) \right)$$

$$= y^\top \prod_{i=0}^{t} \left( I - \frac{1}{n} \sigma \left( X^\top A^{(i)} X \right) \right).$$

Therefore, for $y_q^{(t)}$ we have

$$y_q^{(t+1)} = y_q^{(t)} + \frac{1}{n} y^{(t)^\top} \sigma \left( X^\top A^{(t)} x_q \right)$$

$$= \sum_{j=1}^{t-1} y^\top \left( I - \prod_{i=0}^{j-1} \left( I - \frac{1}{n} \sigma \left( X^\top A^{(i)} X \right) \right) \right) \sigma \left( X^\top A^{(j)} x_q \right) + \prod_{i=0}^{t-1} \left( I - \frac{1}{n} \sigma \left( X^\top A^{(i)} X \right) \right) \sigma \left( X^\top A^{(t)} x_q \right).$$

$\square$

Next, we state a Lemma which is at the core of the proof of Lemmas 1 and Theorem 1.

**Lemma 3** (Polynomial blow up). *Suppose $P(\gamma)$ is an arbitrary polynomial of degree $k$ with $P(0) = 0$. Then, given interval $(\lambda_{min}, \lambda_{max})$ and $k \leq \frac{\sqrt{\lambda_{max}}}{6\sqrt{\lambda_{min}}}$, for an arbitrary choice of $\alpha \in \mathbb{R}$, there exists a choice of $\tilde{\gamma} \in (\lambda_{min}, \lambda_{max})$ such that*

$$\frac{|P(\tilde{\gamma}) - \alpha|}{|\alpha|} \geq \frac{1}{4}.$$

*Furthermore, there exists an interval $(a, b)$ of length at least $\frac{\lambda_{\max} - \lambda_{\min}}{64k^2}$ such that for all $\gamma \in (a, b)$:*

$$\frac{|P(\gamma) - \alpha|}{|\alpha|} \geq \frac{1}{8}.$$

*Proof.* Let $\kappa = \frac{\lambda_{min}}{\lambda_{max}}$. Consider the following scaling of the $k$th Chebyshev polynomial $T_k(\gamma)$:

$$Q_k(\gamma) = T_k\left( \frac{2\gamma - (\lambda_{min} + \lambda_{max})}{\lambda_{max} - \lambda_{min}} \right).$$

Now from the fact that $T_k(x) = (x + \sqrt{x^2 - 1})^k + (x - \sqrt{x^2 - 1})^k$ outside the interval $[-1, 1]$, we get for $x = -1 - \alpha$:

$$(1 + \sqrt{\alpha})^k \leq |T_k(x)| \leq 1 + (1 + 3\sqrt{\alpha})^k.$$

Therefore, for $\kappa = \lambda_{min}/\lambda_{max}$ we have

$$Q_k(0) = T_k\left( 1 - \frac{2\lambda_{min}}{\lambda_{max} - \lambda_{min}} \right) \leq 1 + (1 + 3\sqrt{4\kappa})^k,$$

and using the fact that $k \leq 1/(6\sqrt{\kappa})$, we have $Q_k(0) \leq 4$. On the other hand, from the property of Chebyshev polynomials, we get that $Q_k(0)$ alternates between $-1$ and $1$ $k$ times. This means that if we define $R(\gamma) = Q_k(\gamma)/Q_k(0)$, we have that $R(0) = 1$, and that $R$ alternates above and below the $y = \frac{1}{4}$ and $y = -\frac{1}{4}$ lines $k$ times. Now suppose the absolute value of $P(\gamma)$ is always within the interval $(\alpha - \alpha/4, \alpha + \alpha/4)$. Then for the polynomial $\tilde{P}(\gamma) = 1 - P(\gamma)/\alpha$ we have $\tilde{P}(0) = 1$ and $\tilde{P}(\gamma) < \frac{1}{4}$ for all points $\gamma$ in the interval $[\lambda_{min}, \lambda_{max}]$. Therefore, if we consider the polynomial $S(\gamma) = \tilde{P}(\gamma) - R(\gamma)$, then $S$ has degree at most $k$, $S(0) = 0$, and $S(\gamma)$ alternates $t$ times between positive and negative numbers in the interval $[\lambda_{min}, \lambda_{max}]$. But this means that $S$ has at least $t + 1$ real roots. The contradiction finishes the proof for the first part.

**Proof for the second part.** To show the second part, note that the $k$th Chebyshev polynomial $T_k(x)$

is Lipschitz around its roots; In particular, consider the trigonomic property $T_k(\cos(\theta)) = \cos(k\theta)$, now defining $x(\theta) = \cos(\theta)$, we see that

$$
\begin{aligned}
\frac{d}{dx}T_k(x(\theta)) &= \frac{d}{d\theta}T_k(x(\theta))\frac{d\theta}{dx} \\
&= \frac{d}{d\theta}\cos(k\theta)\left(\frac{dx}{d\theta}\right)^{-1} \\
&= k\sin(k\theta)/\sin(\theta).
\end{aligned}
$$

Namely, the extremal points of $T_K(\cos(\theta))$ are at $\theta = \frac{i\pi}{k}$ for $i = 0, \ldots, k-1$. Now for $|\Delta\theta| \leq \frac{\pi}{4k}$,

$$
T_k(\cos(i\frac{\pi}{k} + \Delta\theta)) = \cos(\pi k + k\Delta) \geq \cos(\frac{\pi}{4}) = \frac{1}{\sqrt{2}} \geq \frac{1}{2}.
$$

Therefore, we conclude that the $i$th root $\cos(\frac{\pi i}{k})$ of $T_k$, for all $x$ in the interval $(\cos(\frac{\pi i}{k} - \frac{\pi}{4k}), \cos(\frac{\pi i}{k} + \frac{\pi}{4k}))$ we have

$$
T_k(x) \geq \frac{1}{2}.
$$

Moreover, it is easy to see that the length of this interval is maximized for $i = 0$ with length $2(1 - \cos(\frac{\pi}{k}))$. Therefore, for all $0 \leq i < k$, if we define the intervals $(a_i, b_i)$

$$
= (\cos(\frac{\pi i}{k} - \frac{\pi}{4k})(\frac{\lambda_{\max} - \lambda_{\min}}{2}) + \frac{\lambda_{\min} + \lambda_{\max}}{2}, \cos(\frac{\pi i}{k} + \frac{\pi}{4k})(\frac{\lambda_{\max} - \lambda_{\min}}{2}) + \frac{\lambda_{\min} + \lambda_{\max}}{2}),
$$

then the scaled Chebyshev polynomials $Q_k$ that we defined for the previous part have the property that for all $0 \leq i < k$ and $\gamma \in (a_i, b_i)$, $Q_k(\gamma) \geq \frac{1}{2}$, and therefore $R(\gamma) \geq \frac{1}{8}$. Now suppose for every $0 \leq i < k$, there exists $\gamma_i \in (a_i, b_i)$ such that $P(\gamma_i) < \frac{1}{8}$. Then, for the polynomial $S(\gamma) = \tilde{P}(\gamma) - R(\gamma)$ we see that again it alternates $t$ times between positive and negative at $\gamma_i$'s, hence has at least $k+1$ roots, which is again a contradiction. Therefore, we conclude that there exists $0 \leq i < k$ such that for all $\gamma \in (a_i, b_i)$:

$$
\frac{|P(\tilde{\gamma}) - \alpha|}{|\alpha|} \geq \frac{1}{8}.
$$

Finally, note that the length of the intervals $(a_i, b_i)$ is minimized for $i = 0$, in which case the length is

$$
(1 - \cos(\frac{\pi}{4k}))(\lambda_{\max} - \lambda_{\min}) \geq \left|1 - \sqrt{1 - \frac{\pi^2}{16k^2}}(\lambda_{\max} - \lambda_{\min})\right| \geq \frac{\pi^2}{64k^2}(\lambda_{\max} - \lambda_{\min}).
$$

$\square$

## C.2 Proof of Lemma 1 and Theorem 1

Proof of Theorem 1 directly follows from Lemma 1.

*Proof of Lemma 1.* First we consider the case when the weight matrices are restricted to $A^{(t)}, u^{(t)}$ as in equation 5. We substitute $X = \gamma U$ and $x_q = \theta x$ with parameters $\gamma, \theta$ and arbitrary fixed matrix $U$ and vector $x$. The key idea is to look at $y_q^{(t)}$ as a polynomial of $\gamma$ and $\theta$, $P_t(\gamma)$, where the degree of $P_t(\gamma)$ is at most $2t$ and $P_t(0) = 0$.

We show this by induction. In particular, we strengthen the argument by also proving that each entry of $y^{(t)}$ is a polynomial of degree at most $2t + 1$ of $\gamma$. The base of induction is clear since $y_q^{(0)} = 0, y^{(0)} = y$ are both degree zero polynomials of $\gamma$. For the step of induction, suppose the argument holds for $t$ and we want to prove it for $t + 1$. We write the update rule for $y_q^{(t)}$:

$$
y_q^{(t+1)} = y_q^{(t)} - \frac{1}{n}(y^{(t)\top} + u^{(t)\top}X)\sigma\left(X^\top A^{(t)}x_q\right). \tag{13}
$$

Now from the hypothesis of induction, we know that $y_q^{(t)}$ is represented by a polynomial of degrees at most $2t$ of $\gamma$. On the other hand, note that from the positive scaling property of ReLU

$$\sigma\left(X^\top A^{(t)} x_q\right) = \sigma\left(\gamma U^\top A^{(t)} x_q\right) = \gamma \sigma\left(U^\top A^{(t)} x_q\right).$$

Therefore, the second part of the second term in equation 13, i.e. $u^{(t)^\top} X \sigma\left(X^\top A^{(t)} x_q\right)$ has degree exactly 2 in $\gamma$. On the other hand, from the hypothesis of induction we know that each entry of $y^{(t)}$ is a polynomial of degree at most $2t + 1$. Therefore, the first part of the second term, $\frac{1}{n} y^{(t)^\top} \sigma\left(X^\top A^{(t)} x_q\right)$, has degree at most $2t + 2$. Hence, overall $y_q^{(t+1)}$ is a polynomial of degree at most $2t + 2$ of $\gamma$. Next, we show the step of induction for $y^{(t+1)}$. We have the update rule

$$y^{(t+1)^\top} = y^{(t)^\top} - \frac{1}{n}(y^{(t)^\top} + u^{(t)^\top} X)\sigma\left(X^\top A^{(t)} X\right).$$

Now again from the scale property of the activation

$$\sigma\left(X^\top A^{(t)} X\right) = \sigma\left((\gamma U)^\top A^{(t)} (\gamma U)\right) = \gamma^2 \sigma\left(U^\top A^{(t)} U\right).$$

Hence, from the hypothesis of induction, $y^{(t)}$ is a polynomial of degree at most $2t + 1$, $y^{(t)^\top} \sigma\left(X^\top A^{(t)} X\right)$ is of degree at most $2t + 3$, and $u^{(t)^\top} X \sigma\left(X^\top A^{(t)} X\right)$ is of degree at most three. Overall we conclude that $y^{(t+1)}$ is of degree at most $2t + 3$ which finishes the step of induction. The result then follows from Lemma 3.

Next, we show the second argument without the weight restriction equation 5 with a similar strategy, namely we substitute $X = \gamma U$ and bound the degree of polynomial that entries of $Z^{(t)}$ are of variable $\gamma$. In particular we claim that each entry of $Z^{(t)}$ is a polynomial of degree at most $3^t$ of $\gamma$. The base of induction is trivial as $Z^{(1)}$ is a polynomial of degree at most 1 of $\gamma$. Now for step of induction, given that $Z^{(t)}$ is a polynomial of degree at most $3^t$, in light of the recursive relation equation 2, $Z^{(t)}$ and the one-homogeneity of ReLU, we get that $Z^{(t+1)}$ is of degree at most $3^{t+1}$, proving the step of induction. The result follows again from Lemma 3. $\square$

### C.3 PROOF OF THEOREM 2

*Proof of Theorem 2.* Using Lemma 4, setting $u^{(i)} = 0, \forall i \leq L$ we have

$$|\text{TF}(Z^{(0)}, \{A^{(t)}, u^{(t)}\}_{t=0}^{L-1}) - w^{*\top} x_q|$$
$$= |y_q - w^{*\top} x_q|$$
$$= |w^{*\top} \prod_{i=0}^{L-1} (I - \Sigma A^{(i)}) x_q|.$$

Now pick $A^{(i)} = \theta_i^{-1} I$, where $\theta_i = \left(\frac{\beta - \alpha}{2} \lambda_i + \frac{\beta + \alpha}{2}\right)^{-1}$ is the $i$th root of the Chebyshev polynomial scaled into the interval $(\alpha, \beta)$. This way

$$\prod_{i=0}^{L-1} (I - \Sigma A^{(i)}) = \prod_{i=0}^{t-1} (I - \Sigma/\theta_i)$$
$$= C \prod_{i=0}^{t-1} (\theta_i I - \Sigma)$$
$$= C P^{L,(\alpha,\beta)}(\Sigma),$$

where $P^{L,(\alpha,\beta)}(x) = T_L(\frac{2x - (\alpha+\beta)}{\beta - \alpha})$ is the degree $L$ Chebyshev polynomial whose roots are scaled into interval $[\alpha, \beta]$ and $C$ is a constant. Note that the constant $C$ is picked in a way that $P^{L,(\alpha,\beta)}(0) = 1$. From , this condition implies that $C \leq \frac{1}{2^L \sqrt{\alpha/\beta} - 1}$, hence picking degree

$L = O(\ln(1/\epsilon)\sqrt{\beta/\alpha})$ implies that the value of $P^{L,(\alpha,\beta)}$ on the interval $[\alpha,\beta]$ is at most $\epsilon$. Now looking at the eigenvalues, this implies

$$-\epsilon I \leq \prod_{i=0}^{t-1}(I - \Sigma/\lambda_i) \leq \epsilon I.$$

Finally from Cauchy-Schwarz

$$\left| w^{*\top} \prod_{i=0}^{L-1}(I - \Sigma A^{(i)})x_q \right|$$

$$\leq \|w^*\|\|x_q\|\|\prod_{i=0}^{L-1}(I - \Sigma A^{(i)})\| \leq \epsilon\|w^*\|\|x_q\|,$$

which completes the proof. $\qquad\square$

## D    ROBUSTNESS OF LOOPED TRANSFORMERS

**Lemma 4** (Transformer and loss formulas). *Transformer output can be calculated as*

$$TF(Z^{(0)}, \{A^{(t)}, u^{(t)}\}_{t=0}^{L-1}) = y^{(L)^\top} = y_q - w^{*\top}\prod_{i=0}^{t-1}(I - \Sigma A^{(i)})x_q.$$

*Furthermore, the loss can be written as*

$$\mathcal{L}(\{A^{(t)}, 0\}_{t=0}^{L-1}) = \mathbb{E}_X trace\left(\prod_{t=0}^{L-1}(I - \Sigma^{1/2}A^{(t)}\Sigma^{1/2})\prod_{t=L-1}^{0}(I - \Sigma^{1/2}A^{(t)}\Sigma^{1/2})\right).$$

*Proof.* First, we show the following recursive relations:

$$y^{(t)^\top} = w^{*\top}\prod_{i=0}^{t-1}(I - \Sigma A^{(i)})X \tag{14}$$

$$y_q^{(t)^\top} = y_q - w^{*\top}\prod_{i=0}^{t-1}(I - \Sigma A^{(i)})x_q, \tag{15}$$

We show this by induction on $t$. For the base case, we have $w^{*\top}X = y^{(0)}$ and $y_q^{(0)} = 0 = y_q - w^{*\top}x_q$. For the step of induction, we have equation 14 and equation 15 for $t - 1$. We then open the following update rule:

$$Z^{(t)} = Z^{(t-1)} - \frac{1}{n}P^{(t)}MZ^{(t)}A^{(t)}Z^{(t)^\top},$$

as

$$y^{(t+1)^\top} = y^{(t)^\top} - \frac{1}{n}y^{(t)^\top}X^\top A^{(t)}X$$

$$= w^{*\top}\prod_{i=0}^{t-1}(I - \Sigma A^{(i)})X - w^{*\top}\prod_{i=0}^{t-1}(I - \Sigma A^{(i)})(XX^\top)A^{(i)}X$$

$$= w^{*\top}\prod_{i=0}^{t}(I - \Sigma A^{(t)})X,$$

where we used the fact that $XX^\top = \Sigma$. Moreover

$$y_q^{(t+1)} = y_q^{(t)} - \frac{1}{n}y^{(t)}{}^\top X^\top Q x_q$$

$$= y_q - w^*{}^\top \prod_{i=0}^{t-1}(I - \Sigma A^{(i)})x_q$$

$$- w^*{}^\top \prod_{i=0}^{t-1}(I - \Sigma A^{(i)})(XX^\top)A^{(i)}x_q$$

$$= y_q - w^*{}^\top \prod_{i=0}^{t}(I - \Sigma A^{(i)})x_q.$$

Therefore

$$\mathbb{E}X(y_q^{(L)} - y_q)^2 = \mathbb{E}X\left(w^*{}^\top \prod_{i=0}^{L-1}(I - \Sigma A^{(i)})x_q\right)^2$$

$$= \mathbb{E}X\left(w^*{}^\top \Sigma^{1/2} \prod_{i=0}^{L-1}(I - \Sigma^{1/2}A^{(i)}\Sigma^{1/2})\Sigma^{-1/2}x_q\right)^2.$$

Now taking expectation with respect to $w^* \sim N(0, \Sigma^{-1})$, $x_q \sim N(0, \Sigma)$:

$$\mathcal{L}(\{A^{(t)}, 0\}_{t=0}^{L-1}) = \mathbb{E}X\text{trace}\left(\prod_{i=0}^{L-1}(I - \Sigma^{1/2}A^{(i)}\Sigma^{1/2})\prod_{i=L-1}^{0}(I - \Sigma^{1/2}A^{(i)}\Sigma^{1/2})\right).$$

$\square$

## D.1 PROOF OF THEOREM 6

*Proof of Theorem 6.* Consider a set of diagonal matrices $\{B^{(t)}\}_{t=0}^{L-1}$ with a set of $Ld$ distinct diagonal entries that satisfy $\alpha I \preccurlyeq B^{(t)}{}^{-1} \preccurlyeq 2\alpha I$ for all $0 \leq t \leq L - 2$ and $B^{(L-1)}{}^{-1} = \beta I$, and define the distribution

$$P_{\alpha,\beta} \triangleq \frac{1}{L}\sum_{t=0}^{L-1}\delta_{B^{(t)}},$$

where $\delta_{B^{(t)}}$ is a point mass on $B^{(t)}$. Moreover, for parameter $\delta'$, consider an alternative distribution $\tilde{P}_{\alpha,\beta}$ on $\Sigma$ where it has at least $\epsilon$ amount of mass on matrices $4\alpha I \preccurlyeq \Sigma \preccurlyeq (1-\delta')\beta I$. First, note that the global minimum of $\mathcal{L}^{P_{\alpha,\beta}}$ is zero and $\mathcal{L}^{P_{\alpha,\beta}}(\{A^{(t)^*}\}_{t=0}^{L-1}, 0)$ for $A^{(t)^*} \triangleq B^{(t)}{}^{-1}\forall 0 \leq t \leq L-1$.

Now for an arbitrary matrix $\Sigma$ such that $8\alpha I \preccurlyeq \Sigma \preccurlyeq (1-\delta')\beta I$, we have for all $0 \leq t \leq L - 2$:

$$\Sigma^{1/2}A^{(t)^*}\Sigma^{1/2} = \Sigma^{1/2}B^{(t)}{}^{-1}\Sigma^{1/2}$$

$$\succcurlyeq \Sigma^{1/2}2\alpha I^{-1}\Sigma^{1/2}$$

$$\succcurlyeq 4I,$$

which implies

$$I - \Sigma^{1/2}A^{(t)^*}\Sigma^{1/2} \preccurlyeq 3I.$$

Therefore, for all $0 \leq t \leq L - 2$,

$$\left(I - \Sigma^{1/2}A^{(t)^*}\Sigma^{1/2}\right)^2 \geq 9I.$$

On the other hand, for $t = L - 1$:

$$\Sigma^{1/2}A^{(L-1)^*}\Sigma^{1/2} = \frac{1}{\beta}\Sigma \leq (1-\delta')I,$$

which implies

$$I - \Sigma^{1/2}A^{(L-1)^*}\Sigma^{1/2} \succcurlyeq \delta'I.$$

Hence,

$$\left(I - \Sigma^{1/2}A^{(L-1)^*}\Sigma^{1/2}\right)^2 \geq \delta'^2 I.$$

Therefore, by repeating the inequality $CBC^\top \geq CDC^\top$ for arbitrary matrices $B, C, D$ where $B \succcurlyeq D$, we get

$$\prod_{t=0}^{L-1}(I - \Sigma^{1/2}A^{(t)^*}\Sigma^{1/2}) \prod_{t=L-1}^{0}(I - \Sigma^{1/2}A^{(t)^*}\Sigma^{1/2}) \succcurlyeq \delta'9^{L-1}I.$$

Therefore

$$\mathrm{trace}\left(\prod_{t=0}^{L-1}(I - \Sigma^{1/2}A^{(t)^*}\Sigma^{1/2}) \prod_{t=L-1}^{0}(I - \Sigma^{1/2}A^{(t)^*}\Sigma^{1/2})\right) \geq d\delta'9^{L-1}.$$

On the other hand, note that from our assumption the distribution $\tilde{P}_{\alpha,\beta}$ has at least $\epsilon$ mass on matrices $\Sigma$ with $4\alpha I \preccurlyeq \Sigma \preccurlyeq (1-\delta')\beta I$. Therefore

$$\mathcal{L}^{\tilde{P}_{\alpha,\beta}}(\{A^{(t)^*}\}_{t=0}^{L-1}, 0) \geq \epsilon\delta' d9^{L-1}.$$

$\square$

# E    PROOF OF THEOREM 3

*Proof.* From Lemma 4 (for $u^{(t)} = 0$), $\forall t \leq L$, we have

$$y^{(t)^\top} = w^{*\top}\prod_{i=0}^{t-1}(I - \Sigma A^{(i)})X,$$

$$y_q^{(t)} = y_q - w^{*\top}\prod_{i=0}^{t-1}(I - \Sigma A^{(i)})x_q.$$

Therefore

$$\|y^{(L)}\|^2 = \|X^\top\prod_{i=0}^{L-1}(I - \Sigma A^{(i)})w^*\|^2$$

$$= \|\prod_{i=0}^{L-1}(I - \Sigma A^{(i)})w^*\|_{XX^\top}^2.$$

Therefore,

$$|y_q - y_q^{(L)}| = |w^{*\top}\prod_{i=0}^{L-1}(I - \Sigma A^{(i)})x_q|$$

$$\leq \|x_q\|_{(XX^\top)^{-1}}\|\prod_{i=0}^{L-1}(I - \Sigma A^{(i)})w^*\|_{XX^\top}$$

$$\leq \epsilon.$$

$\square$

# F PROOF OF THEOREM 8

*Proof of Theorem 8.* Suppose $\Sigma^{1/2} = vv^\top$ is rank one and define
$$v^\top A^{(i)} v \|v\|^2 = \gamma_i.$$
Then
$$\Sigma^{1/2} A^{(i)} \Sigma^{1/2} = vv^\top A^{(i)} vv^\top = \gamma_i \tilde{v}\tilde{v}^\top,$$
for $\gamma_i > 2$, where $\tilde{v}$ is the normalized version of $v$. This implies
$$\left(I - \Sigma^{1/2} A^{(i)} \Sigma^{1/2}\right) = (1 - \gamma_i)\tilde{v}\tilde{v}^\top + \left(I - \tilde{v}\tilde{v}^\top\right),$$
which then gives
$$\mathcal{L}^\Sigma(\{A^{(i)}\}_{i=0}^L, 0) = \text{trace}\left(\prod_{i=0}^{L-1}(I - \Sigma^{1/2} A^{(i)} \Sigma^{1/2}) \prod_{i=L-1}^{0}(I - \Sigma^{1/2} A^{(i)} \Sigma^{1/2})\right)$$
$$= \prod_{i=0}^{L-1}(\gamma_i - 1)^2 + (d - 1).$$
Therefore, the loss is increasing with depth if $\gamma_t > 2$ and decreasing if $\gamma_t < 2$. Therefore, for the behavior of the loss wrt to depth to be either increasing or decreasing for all $t$, for every $v$, we need to have that either all $\gamma_i$'s are great than or equal to 2, or less than or equal to 2. But this implies that for all $v$ and all $i, j$:
$$v^\top A^{(i)} v = v^\top A^{(j)} v.$$
Suppose this is not the case for some $v, i, j$, namely
$$v^\top A^{(i)} v < v^\top A^{(j)} v.$$
Then, we can scale $v$ so that $v^\top A^{(i)} v < 2$ and $v^\top A^{(i)} v > 2$ which contradicts the monotonicity of the loss wrt to depth. Therefore, for all $v$, and all $i, j$, we showed
$$v^\top A^{(i)} v = v^\top A^{(j)} v,$$
which implies for all $i, j \in [L]$ we should have
$$A^{(i)} = A^{(j)}.$$

Next, we show the argument for loop models. Namely, suppose $A^{(1)} = \cdots = A^{(L)} = A$. The loss in this case becomes
$$\mathcal{L}^\Sigma(\{A\}_{i=0}^L, 0) = \text{trace}\left((I - \Sigma^{1/2} A \Sigma^{1/2})^{2L}\right). \tag{16}$$
Now let $\beta_1 < \cdots < \beta_{d-1}$ are the eigenvalues of $\Sigma^{1/2} A \Sigma^{1/2}$. Then
$$\mathcal{L}^\Sigma(\{A\}_{i=0}^L, 0) = \sum_{i=0}^{d_1}(1 - \beta_i)^{2L}. \tag{17}$$
Now if $|1 - \beta_i| \leq 1$ for all $0 \leq i \leq d - 1$, then each term $(1 - \beta_i)^{2L}$ is decreasing in $L$. Otherwise, suppose there is $j$ such that $|1 - \beta_j| > 1$. Then, defining $L_0 = \ln(d/(|1 - \beta_j| - 1))/\ln(|1 - \beta_j|)$, then for $L \geq L_0$, we have
$$(1 - \beta_j)^{2L} \geq \frac{d}{|1 - \beta_j| - 1}.$$
Hence,
$$(1 - \beta_j)^{2(L+1)} \geq (1 - \beta_j)^{2L} + (1 - \beta_j)^{2L} \times (|1 - \beta_j| - 1) \tag{18}$$
$$\geq (1 - \beta_j)^{2L} + d. \tag{19}$$
This means going from $L$ to $L + 1$, the loss increase by at least $d$, while the potential decrease in the loss from the other $d - 1$ terms in equation 17 is at most $d - 1$. Hence, we showed that for all $L \geq L_0$, the loss is increasing in this case, which compeletes the proof.

$\square$

