# OpenReview forum: "On the Role of Depth and Looping for In-Context Learning with Task Diversity"
_ICLR.cc/2025/Conference — Submitted to ICLR 2025_

### Official Review · Reviewer_PS8E · 2024-10-21

**Soundness:** 3
**Presentation:** 2
**Contribution:** 2
**Rating:** 5
**Confidence:** 4

**Summary:**

The paper studies the in-context learning (ICL) ability of transformers consisting of multiple attention layers to learn linear regression tasks with diverse covariances where the eigenvalues are assumed to lie in $[\alpha,\beta]$. A lower bound of $\Omega(\sqrt{\beta/\alpha})$ for the constrained case and $\Omega(\log\beta/\alpha)$ for the unconstrained case on the number of layers required to solve all such tasks are established, and transformer constructions achieving matching upper bounds are provided. It is also shown that multilayer Transformers are not robust to task distribution shifts due to overfitting, while looped Transformers can generalize to any distribution, showing that looped models achieve both expressivity and robustness.

**Strengths:**

The task-diverse regression setting constitutes a nice extension to the existing body of literature on ICL of linear regression where learnability can be characterized in terms of the condition number $\beta/\alpha$. The provided upper and lower bounds match, providing a tight characterization of the expressivity of multilayer Transformers in this setting.

**Weaknesses:**

* The architecture under study is quite stylized: the attention mechanism is restricted to be linear or ReLU (robustness results only to apply to the linear case) and feedforward layers or other operations are not considered. While such restrictions are common in the ICL literature, I think this is a significant limitation for this particular paper since the main novelty seems to be the lower bounds. To be specific, the upper bounds in Theorems 4 and 5 (essentially established in previous works) tell us that practical Transformers are capable of executing a variety algorithms in context, even with certain architectural restrictions, and many derivative works have already explored this correspondence. However, the newly provided lower bounds do not shed much light on the limitations of practical Transformers, especially since the proof relies on a polynomial argument which is very 'brittle' and does not give any insight if a non-polynomial aspect such as softmax attention or feedforward layers is added.
* The above weakness could be mitigated via numerical experiments which demonstrate similar deficiencies in softmax Transformers. However, the experiments are limited to restricted linear attention.
* The strong out-of-distribution capabilities of looped Transformers have already been theoretically established in [1]. Their Theorem 4.4 shows that (quote) "a looped Transformer learned on one in-context distribution can generalize to other problem instances with different covariance, owing to the fact that it has learned a good iterative algorithm."
* Unlike [1], the paper does not provide any optimization guarantees for either looped or ordinary Transformers.
* The writing of the paper could be improved. In particular, the Appendix needs to be better structured: the ordering of the proofs does not match the results in the main text, some proofs refer to lemmas that are placed later in the Appendix, and there is often no indication in the main text where the corresponding proofs are, making the logic hard to follow. There are also frequent typos throughout the paper, see below.

**Typos and errors**

* Theorem 1: the second part should refer to non-restricted rather than restricted attention. The bound $L\le \sqrt{\beta/\alpha}$ should be $O(\sqrt{\beta/\alpha})$. Equation (7) should not have the $\gamma$ factor.
* Lemma 1 has no proof and having both Lemma 1 and Theorem 1 in the main body of text is redundant. Capitalization of "1. there" should be fixed.
* Theorem 4: The statement of Newton's method is wrong.
* Theorem 7: $P(\alpha,\beta)$ needs to be $P_{\alpha,\beta}$.
* Proof of "Lemma 7": needs to be Theorem 7. In the same paragraph, $A$ needs to be defined first and Lemma 4 should come before this.
* The next sentence reads "we get $I-\beta^{-1}\Sigma\le I$, hence $(I-\beta^{-1}\Sigma)\le I$"
* $1-\delta'$ instead of $1\delta'$ after (12)
* The sentence before Lemma 3 does not make sense and should be deleted.
* Need clarification: what is $\lambda_{min},\lambda_{max}$ in Lemma 3?
* In the proof of Lemma 3: lipschitz, chebyshev (need capitalization); miaximized, conlcude, taht, legnth (typos)
* Capitalization such as looped/Looped, equation/Equation is inconsistent.
* There are many instances of "Equation equation X" which need to be fixed.
* There are many instances of $\mathbb{E}_{XYZ}$ being displayed as $\mathbb{E}XYZ$.

[1] Gatmiry et al. Can Looped Transformers Learn to Implement Multi-step Gradient Descent for In-context Learning? ICML 2024.

**Questions:**

* Is there any way for the results or techniques to be extended to, or suggest new directions regarding, softmax attention or nonlinear feedforward layers?
* The lower bound for restricted attention is not matched by looped Transformers in Theorem 5. Is there an expressivity gap in this setting?
* Regarding out-of-distribution learning capability, what is the main difference with the result in [1]? Is the reason for robustness due to learning a good iterative algorithm as [1] suggests?

---

> ### Author Response · Authors · 2024-11-26
> **Response to reviewer PS8E**
>
> We thank the reviewer for their feedback and pointing out to our typos. Below we have address all their concerns. We have also applied all their comments to the new version of the paper uploaded. In addition, we believe there are some misunderstandings on our contribution and how in compares to the literature that we will clarify below.
>
> **Relu in identity for activation is restricted and does not cover all cases that are interesting in practice, and what about non-polynomial activations?**
>
> **Reply:** First we note that Relu is a non-polynomial activation. In fact our lower bounding method works after applying feedforward layers or other non-linearities than Relu by approximating the activations with polynomials**. In other words, if the approximation degree is not superlarge, the results are generalizable, but for sake of keeping the presentation of the work simple and avoid generalizations that lack deep novel ideas and follows from more conventional derivations, we decided not to add this sort of generalizations to our current work (Since we have many results including depth lower and upper bounds and robustness aspects etc in the current work). We point out that our work is the first attempt to show any such expressivity lower bound (i.e. limitations) for in-context learning, since most previous works focussed on showing upper bounds for Transformers. We hope this work encourages more study on the limitations of Transformers and the importance of depth in more general settings.
>
> **The above weakness could be mitigated via numerical experiments**
>
> **Reply:** Our theoretical work was, in fact, motivated by previous empirical works by Garg et al. and Yang et al. which show that Transformers with softmax activations can solve ICL problems with sufficient depth. Since our theoretical analysis was in a slightly different setting, we ran experiments in the restricted linear setting to verify that the findings also hold there.
>
> **Out of distribution generalization was studied in Gatmiry et al [1]**
>
> **Reply:** Note that the out-of-distribution result in Gatmiry only holds for the global minimizer of the Gaussian case, i.e. when data is sampled iid from a single Gaussian, in which case the data covariance distribution is concentrated on the covariance of the Gaussian distribution. We significantly generalize that result to a much broader class of distributions over the covariance, namely the class of right-spread-out distributions, which can be highly multi-modal. We believe that this is in some sense the minimal assumption on the training distribution that enables one to obtain out-of-distribution generalization.
>
>
> **The paper does not provide an optimization guarantee**
>
> **Reply:** The focus of this work is on expressivity of multi-layer and looped transformers and analyzing the global minimizer of distributions that are much more general than Gaussian, and their connections with robustness and out-of-distribution generalization. We believe that we have proved many interesting results in this work around these topics that have to be self-contained in this paper. Proving optimization results is certainly an interesting future direction.
>
>
> **The writing of the paper can be improved**
>
> **Reply:** We have applied all the comments of reviewers and improved the presentation of the work in the new version that we have uploaded. Given that all these concerns are addressed, we kindly ask the reviewer to reconsider their score.
>
> **Lemma 1 in the main body seems unnecessary**
>
> **Reply:** In fact we see Lemma 1 as the core technical result and Theorem 1 as its corollary. Hence, we decided to put Lemma 1 in the main body to just highlight our approach for proving the lower bound in Theorem 1. That being said, the proof in the appendix is really the proof of Lemma 1. We made this point clearer in the new version. Thanks for your feedback.
>
> **We also applied all the feedback on typos and minor errors that the reviewer kindly provided to the new uploaded version.**
>
> **Questions**
>
> **Is there a way to generalize the techniques in this paper to softmax attention or non-linear feedforward networks?**
>
> **Reply:** Please see our answer to your first point above
>
> **The lower bound for restricted attention is not matched by upper bound Looped transformer, is there a gap in expressivity here?**
>
> **Reply:** That is a great point, although we have not shown any formal result here, we suspect that in fact the loop model is also capable of achieving the $\sqrt{\beta/\alpha}$ complexity by a more complicated construction. We leave this interesting direction for future work.
>
> **What is the difference between the out-of-distribution upper bound in your work and that of Gatmiry et al. [1]**
>
> **Reply:** Please see our answer to your second point above.

---

> > ### Comment · Reviewer_PS8E · 2024-11-29
> >
> > Thank you for the reply. In the first point, for the polynomial argument I meant polynomial in the scaling factor $\gamma$ (due to the homogeneity of ReLU), not the activation function. Nonetheless, if the lower bound indeed works for general activations as the authors mention, the result is indeed quite interesting. I will raise my score to 5 but ask the authors to include a section or remark in the paper discussing the approximation argument in a bit more detail; I am interested in seeing how the lower bounds differ in that case.

---

### Official Review · Reviewer_wVSr · 2024-10-26

**Soundness:** 4
**Presentation:** 3
**Contribution:** 3
**Rating:** 8
**Confidence:** 3

**Summary:**

This paper follows a line of work studying the ability of transformers to learn to solve linear regression problems in-context. The key novelty is the consideration of the training tasks having condition number within an interval. The paper theoretically examines transformers with ReLU- and identity-activated attention (as opposed to softmax) and without an MLP or layer normalization, and shows tight upper and lower bounds on the number of layers required to solve tasks with condition number within this interval. The paper also shows that minimizers of the training loss do not generalize to new tasks whose condition number lies outside the interval for the training tasks, whereas a simple and previously-studied modification of the transformer, namely the looped transformer, which repeats attention parameters across layers, indeed generalizes. Toy experiments are conducted to support the theory.

**Strengths:**

- The paper addresses an area of interest to the theoretical community -- understanding the ability of (simplified) transformers to solve linear regression in-context -- and provides novel, decently surprising results on this topic. As the authors point out, considering the case in which the transformers are trained on tasks with varying condition number is an improvement over prior work. The separation between multilayer and looped transformers on OOD data is especially interesting.

- The theory is rigorous, comprehensive, and clearly discussed.

- The paper is very well-written overall.

- The experiments support the theory.

**Weaknesses:**

- The results are limited to simplified transformers that use ReLU or identity activation instead of softmax in the attention block, and do not use MLPs or layer normalization. This is common in the literature, but should be acknowledged in the introduction and abstract.

- Although it has been used previously in the literature, the assumption in the OOD case that $w^*$ and $x_q$ have inverse covariances is quite contrived and limited.

- The experiments are apparently only run once -- they should be repeated over several trials with error bars shown, especially since the curves seem to have large variance, e.g. it is strange that looped transformers generalize worse than multilayer transformers in part of Figure 2b, this could be an outlying trial. Running more trials is thus important for our understanding.

- (Minor) Figure 1a should be log-scaled.

**Questions:**

n/a

---

> ### Author Response · Authors · 2024-11-26
> **Response to Reviewer wVSr**
>
> We thank the reviewer for acknowledging our contributions and their valuable feedback.
>
> **The result does not apply to softmax, multi-layer perceptron networks, etc**
>
> **Reply:** our lower bounding method works after applying feedforward layers or other non-linearities than Relu by approximating the activations with polynomials**. In other words, if the approximation degree is not superlarge, the results are generalizable, but for sake of keeping the presentation of the work simple and avoid generalizations that lack deep novel ideas and follows from more conventional derivations, we decided not to add this sort of generalizations to our current work (Since we have many results including depth lower and upper bounds and robustness aspects etc in the current work). We point out that our work is the first attempt to show any such expressivity lower bound (i.e. limitations) for in-context learning, since most previous works focussed on showing upper bounds for Transformers. We hope this work encourages more study on the limitations of Transformers and the importance of depth in more general settings.
>
>
> **The assumption on the distribution of $w^*$ and $x_q$ although used in previous work is rather contrived*
>
> **Reply:**
> The choice of distribution on $x_q$ is so that it aligns with the distribution of $x_i$’s, since the data covariance matrix defined based on $x_i$’s is going to concentrate around the population covariance. The choice of the distribution of $x^*$ is rather a simplifying assumption for our analysis, and also natural makes the distribution of the output ${w^*}^\top x$ becomes a subexponential whose variance does not depend on $\Sigma$.
>
> **The experiments are apparently only run once and there might be some outliers**
>
> **Reply:** Each experiment has been run 3 times and the results presented are an average over the 3 runs.

---

> > ### Author Response · Authors · 2024-12-02
> > **Following up with Reviewer wVSr**
> >
> > We are following up to see if you had a chance to review our response to your comments/concerns and if you have any further questions. We uploaded a revision by fixing some typos and adding clarifications based on the comments. Thank you for your time and effort in reviewing the submission

---

### Official Review · Reviewer_g1xd · 2024-11-03

**Soundness:** 2
**Presentation:** 1
**Contribution:** 2
**Rating:** 3
**Confidence:** 3

**Summary:**

This paper examines the expressivity of multi-layer ReLU or linear transformers in performing in-context learning (ICL) for linear regression tasks with varying condition numbers (from $1$ to $\kappa$). The authors derive the necessary and sufficient number of layers required for the transformers to solve the ICL problem as a function of $\kappa$. Additionally, the paper compares the out-of-distribution performance of a standard transformer with that of a looped transformer, where the QKV matrices are shared.

**Strengths:**

There might be some merits in the results. But the writing of the paper is so below standard that I am unsure if I understand those results correctly. Therefore I am unwilling to make potentially incorrect assertions about the strength of the paper.

**Weaknesses:**

I find the writing of the paper to be significantly below standard, to the point where it impairs the clarity and understanding of the main results. Below, I outline some potential issues (though I believe there are likely more) and my questions:

- **Line 153:** Based on the context, it should be $X^\top w^* = y$ since $X$ is $d \times n$.
- **Line 212:** According to equation (1), some index $t$ in equation (2) should be $t-1$.
- **Lines 223–228:** The formula in equation (5) is unusual, particularly with the inclusion of $(u^{(t)}, 1)$ in the last row and many zeros in $P^{(t)}$. I could not locate this formula in Ahn et al. 2024a—could you point out where this setting is considered in Ahn et al. 2024a? Additionally, what is the role of the non-zero $u^{(t)}$?
- **Line 267:** In the definition of $Z^{(0)}$, should one of the zeros be $x_q$?
- **Equation (6) in Lemma 1:** I find this equation difficult to interpret. The input to the transformer is rescaled by $\gamma$, which implies that $x_q$ in the input is also rescaled by $\gamma$ (as per the previous question). However, the “label” $(w^*)^\top x_q$ is not rescaled by $\gamma$. If my interpretation is correct, this lemma seems trivial since the transformer output and the label are not on the same scale. I have a similar concern with Theorem 1.
- **Line 281:** $w$ should be $w^*$.
- **Line 284:** “…under restricted attention…”—is this a typo?
- **Theorem 2:** The $\sqrt{\kappa} = \sqrt{\beta/\alpha}$ dependence in the number of layers suggests that the transformers are implementing accelerated gradient descent (AGD) rather than gradient descent (GD). I believe this point is worth a more detailed discussion.
- **Line 350:** Should $k$ be $L$?
- **Section 4.2:** This section is particularly difficult to follow. What is meant by "in-distribution" here? Without first clarifying this, it is hard to grasp what "out-of-distribution" refers to. I also cannot figure out what would be the “(supervised) learning tasks” and “population distribution” in this context, given that $X$ is supposed to be fixed while $x_q$ is random. Moreover, it appears that the transformer only sees one context example during training. If that is the case, how can we expect it to generalize?

The appendix is even more problematic. I only skimmed it but found it nearly unreadable.

- **Line 884:** The sentence is incomplete.
- **Line 1017:** $\lambda_i$ should be $\theta_i$.

In its current state, the submission requires substantial revision and polish. I recommend rejecting the paper in its present form, but I encourage the authors to resubmit after addressing these issues and thoroughly refining the manuscript.

**Questions:**

See above please.

---

> ### Author Response · Authors · 2024-11-26
> **Response to Reviewer g1xd**
>
> We thank the reviewer for their feedback. Below we will clarify the ambiguities.
>
> Line 153: $X w^*$ has to be $X^{\top} w^*$... The index $t$ of $Z$ in line 212 has to be $t+1$... $x_q$ is missing in the definition of $Z^{(0)}$ in line 267
>
> **Reply:** Thank you for pointing out this typo, we fixed it in the new version.
>
>
> **Line 223-228: this is unusual and I could not locate this formula in Ahn et al 2024a**
> **Reply:** Please see Equation (8) in Ahn et al, right before Lemma 1. So our definition is in fact more general than theirs in that we allow non-zero $u^{(t)}$ while in Ahn et al. assume $u^{(t)} = 0$.
>
> **The matrix $Z^{(0)}$ is rescaled in Equation 6 by $\gamma$, which means $x_q$ is also rescaled, which is not the case in the line before.**
>
> **Reply:** Sorry for the confusion, in the current version of the proof we only rescale $X$ and $y$ by $\gamma$ as we have denoted in line 262 (so $x_q$ is not scaled). That being said, the $\gamma$ behind $Z^{(0)}$ in Equation (6) is a typo. Thank you very much for pointing this out, we fixed this typo in the new version uploaded.
>
> **line 281: w^* should be w:**
>
> **Reply:** thank you for spotting this typo. We fixed this typo in the new version.
>
> **line 284: “under restricted attention” is typo?**
>
> **Reply:**Yes, this is a typo, we fixed it in the new version.
>
> **Theorem 2. $\sqrt k$ dependency of the depth is a type of acceleration that is worthy of more discussion**
>
> **Reply:** The reviewer is correct, this is indeed a type of acceleration that is achieved using Chebyshev polynomials. We have added a note about its connection with accelerated rates in optimization within the final version (last sentence before section 3.3).
>
> **Line 350, k is L ?**
>
> **Reply:** Yes thank you for pointing out to this typo, we fixed it in the new uploaded version.
>
> **What is meant by "in-distribution" here? Without first clarifying this, it is hard to grasp what "out-of-distribution" refers to. I also cannot figure out what would be the “(supervised) learning tasks” and “population distribution” in this context, given that $X$ is supposed to be fixed while $x_q$  is random.**
>
> **Reply:** We understand why the reviewer is confused here. To be superclear, in this paper we study the in-context learning problem on the supervised learning task of linear regression instance; so each instance here is a “learning problem” following the formalization from Garg et al. We consider a “population distributions” on these linear regression instances by assuming a distribution on the data covariance matrix $\Sigma$, the query vector $x_q$ and the regressor $w^*$. This is very similar to the assumptions made in prior work for ICL, except that the inputs do not need to have a Gaussian distribution. Following Gatmiry et al [1], we also assume $w\sim \mathcal N(0,\Sigma^{-1})$.
>
> We made changes in the new version in lines 150, 154, and lines 369 and 375 of section 4.2 to make these points clearer. Hopefully this has clarified the confusion for the reviewer.
>
> We also resolved the typos in line 884 and 1017 in the new version.
>
> **Moreover, it appears that the transformer only sees one context example during training. If that is the case, how can we expect it to generalize?**
>
> No, the transformer can see a finite number of samples. Just like any supervised learning problem, even though the model might only see a finite number of samples from a distribution, it can learn to generalize to the population distribution. The situation for ICL should not be any different.

---

### Official Review · Reviewer_6ZLH · 2024-11-04

**Soundness:** 3
**Presentation:** 2
**Contribution:** 2
**Rating:** 5
**Confidence:** 3

**Summary:**

The study explores the in-context learning (ICL) capabilities of deep Transformer models for linear regression tasks with diverse data covariance conditions. It highlights the significance of model depth, showing that multilayer Transformers can solve such tasks with a number of layers matching theoretical lower bounds. Furthermore, the authors show that this expressivity comes at the cost of robustness,  these models are sensitive to minor distributional shifts. The research also studies the Looped Transformers and shows that such a simplified transformer model can be more robust to the distributional shift.

**Strengths:**

The strengths of this paper are summarized as follows:

* This paper considers task-diverse in-context linear regression where the in-context data may have different covariance matrices. Then, this paper proves the lower bound on the number of required attention heads to handle the task-diverse tasks.
* This paper also develops the upper bound on a class of transformer models, which matches the lower bound results.
* This paper also investigates the robustness of the multilayer transformer regarding the distributional shift and reveals its connection with the model depth, and the structural of the transformer (e.g., looped transformer vs. standard transformer).

**Weaknesses:**

The weaknesses of this paper are summarized as follows:

* The presentation of this paper needs improvements. In particular, in Theorem 1, the role of $gamma$ is not clear to me. Why does one need to involve $\gamma$ in this theorem, is there any condition for $\gamma$ (may be the same condition in Lemma 1)?

* In Theorem 1, the statement is made on one particular instance $(X, w^*, y, x_q)$, it is indeed possible that the model cannot provide accurate prediction for all instances. However, in practice one may expect the model to have good performance in expectation or with high probability, would it be possible to extend the lower bound results to the expectation or high probability cases?

* It is not to call "matching upper and lower bound" as the upper bound has an additional $\log(1/\epsilon)$ factor. A more rigorous claim should be "the upper bound matches the lower one up to logarithmic factors."

* Theorem 6 is a bit weird. It claims that there exists a global minimizer that will have very bad robustness as $L$ increases. I just do not quite understand why this argument is made on one existing global minimizer, is it possible that for other global minimizers, the robustness can be better? This should be made very clear.

* The proof is extremely not well organized. Many proofs do not have clean logic and are very hard to follow, thus making it hard to rigorously check the correctness of the proof. For instance, in Lemma 3, does the result hold for any polynomial function $P(\gamma)$?

**Questions:**

Please see the weakness section.

---

> ### Author Response · Authors · 2024-11-26
> **Response to reviewer 6ZLH**
>
> We thank the reviewer for valuable comments and feedback.
>
> **W1: The presentation has to become more clear…. The role of $\gamma$ in Theorem 1 is not clear**
>
> **Reply:** Thank you for your feedback, we try to add more high level discussion to our rigorous proofs to add more intuition. Indeed $\gamma$ has to be removed from Theorem 1.
>
>  **W2: The lower bound is proved for one instance from the class of covariances with bounded condition number. Is it possible to show lower bounds against a distribution, in expectation or high probability?**
>
> **Reply:**
> No, the result is proved against a range of instances whose covariance has bounded condition number.
> To show a lower bound, one considers a “normal” class of instances and prove that given any transformer, there is a bad instance in that class on which the transformer behvaes poor, which is exactly what are result states. The reason is, for any fixed instance, and likely for many fixed distributions over instances, one can find a transformer that achieves low loss on that particular instance or distribution, but the transformer does not necessarily behave well for other instances or distributions.
>
> **W3: It is not to call matching upper and lower bound as the upper bound has an additional $\og(1/\epsilon)$ factor**
>
> **Reply:** Thank you for your feedback on phrasing this part; when mentioning matching upper and lower bound, we mean the upper bound on depth for achieving a constant error, say 0.001, in which the log term is just a constant. We have clarified this in the new version.
>
> **W4: Theorem 6 claims the existence of a global minimzier in which out-of-distribution loss exponentially blows up, what about other global minimizers?**
>
> **Reply:** Thank you for your feedback on this statement. While we suspect that this is the case for all global minimizers (if any other one exists), we currently do not know how to rule out the existence of another minimizer in which the loss might not blow up, as the corresponding matrix equation is highly complex. That being said, existence of bad global minima on its own shows that training transformers with first-order gradient method can lead to these bad solutions that cannot generalize out of distribution. We have added our conjecture about the global minimizer of the loss right after Theorem 6.
>
> **W5: Some parts of proofs are not clear. For example Lemma 3, does the result hold for any polynomial $P(\gamma)$?**
>
> **Reply:** Yes, as we state in the statement of Lemma 3, the result holds for any polynomial $P$ with degree $k$. We have added the adjective “arbitrary” before the word “polynomial” to make this clearer.  While we appreciate the reviewer’s feedback on clarity of the proof and will try to add more details in the final version, we would like to point out that our vision for presenting the proofs was not to make them unnecessarily lengthy and complex, but rather keep them as short and sharp as possible, conveying the key mathematical ideas. If the reviewer can explicitly point out parts that they think are unclear, that would help us a lot in adding more clarity.

---

### Official Review · Reviewer_2duX · 2024-11-07

**Soundness:** 3
**Presentation:** 3
**Contribution:** 3
**Rating:** 6
**Confidence:** 4

**Summary:**

This paper studies in-context learning (ICL) for multiple regression tasks, focusing on "task diversity" by varying the covariances of linear regression tasks. As far as I understood this corresponds to the standard ICL setting where all n tuples (x_i,y_i) are sampled from the same "task", but the model has to be robust to changes of the covariance of the gaussian used to sample x_i. The authors present a set of theoretical results to show that 1) Transformers can do this with depth 2) they are not robust in out of distribution tasks (eg there's a price to pay for depth) 3) looped transformers (eg TFs with weight sharing) are robust to this issue (nice!). The work introduces theoretical lower bounds on the depth of Transformers required for such tasks, and also constructive proofs on how these are matched. It's interesting to see lower bounds in this context, as they are as far as I am aware, pretty rare. Some of their results are supported by experimental evidence.

**Strengths:**

+ The paper establishes architectural lower bounds for ICL problems, which is interesting and relatively rare in this literature.
+ It highlights the lack of robustness in out-of-distribution generalization for standard*  multilayer Transformers, which is valuable.
+ Looped Transformers show promise as they are robust to these issues, aligning well with previous work.
+ Basic experiments support the theoretical claims, adding some evidence to back up the theory.

*(caveating this that the activations are linear/relu)

**Weaknesses:**

- It’s unclear how well these controlled settings will transfer to real-world problems. This is a broader issue in the literature on simple in-context learning (ICL) tasks where x is gaussian and y a regressor label, not just this paper.

- The idea of “diversity” here might be somewhat questionable; using the same task (linear regression) but with different covariances could still mean we’re looking at effectively the same task?

- Choosing linear/ReLU activation on the self-attention matrix may not map well to reality.

- The authors mention that there is “signal” from linear attention for understanding non-linear attention (citing Ahn et al., 2024b). This is vague, and it’s unclear how the signal carries over from linear to standard attention, especially for language tasks. It seems like an added leap of faith without much support. (eg non-language ICL on linear transformer -> non-language ICL on standard transformer -> language ICL on linear transformer). I get that the theory becomes really complicated with other activations, but the amount of relevance of these results to real settings is still unclear.

**Questions:**

- How do the authors justify that linear regression with different covariances sufficiently captures "task diversity"?

- Can the authors clarify the signal that linear attention supposedly provides for non-linear tasks, and the feasibility of extending these findings from non-language to language models?

- This is a "harder" question, and you do not have to answer it for me to increase my score, but a good one to ponder on: In what ways do these results change/further inform the way that a practitioner thinks about transformers and ICL?

---

> ### Author Response · Authors · 2024-11-26
> **Response to Reviewer 2duX**
>
> We thank the reviewer for a detailed review of the paper and for appreciating the contributions like lower bounds and robustness results.
>
> **W1: It’s unclear how well these controlled settings will transfer to real-world problems. This is a broader issue in the literature**
>
> **Reply:**We would like to note that most of the results in the paper do not make the Gaussian assumption. In fact our matching upper and lower bounds consider arbitrary linear regression instances with bounded eigenvalues of the covariance. Moreover, our robustness result for looped models holds for a much broader class of distributions (that satisfy the “right-spread-out” assumption) rather than just a single Gaussian as in previous work.
>
> **W2 The idea of “diversity” here might be somewhat questionable**
>
> **Reply:**We model diversity as different instances that belong to the same learning task, here the learning task is linear regression. While not perfect, this already provides a reasonable test-bed for modeling in-context learning ability. Proving any theoretical result encompassing various learning tasks, although interesting as the reviewer has mentioned, is currently beyond our reach. We believe many fundamental questions about in-context learning of transformers first need to be addressed before that.
>
> **W3: Choosing linear/ReLU activation on the self-attention matrix may not map well to reality.**
>
> **Reply:** We point out that previous theoretical work on in-context learning mainly considered either linear attention, or at most Relu attention (what references are good?). We do acknowledge that the goal is to move to a stronger model, however proving lower bounds becomes even more difficult.
>
> **W4: The authors mention that there is “signal” from linear attention for understanding non-linear attention (citing Ahn et al., 2024b). this is vague**
>
> **Reply:** We acknowledge that indeed the behavior of linear transformers can be different, depending on the specific setting and instances, to that of non-linear attention. The reason behind citing the work of Ahn. et al is that they show linear transformers generalize as well as non-linear ones for many complex regimes, so theoretical understanding of linear transformers is also very valuable. Moreover we note that our lower bounds also apply to attentions with Relu activation. Thus theoretically we have shown that adding Relu non-linearity cannot qualitatively improve the performance of linear transformers, which aligns with the empirical observation of Ahn. et al.
>
> **Q1: How do the authors justify that linear regression with different covariances sufficiently captures "task diversity"?**
>
> **Reply:** The notion of diversity we consider (arbitrary covariances with bounded eigenvalues) is a significant improvement over the settings considered in prior work (unimodal Gaussians). Crucially, for the unimodal Gaussian distribution even a 1-layer model can almost perfectly solve the task. However for the diversity we consider, depth becomes necessary in terms of expressivity as per our results, which makes the setting more appealing.
>
> **Q2: Can the authors clarify the signal that linear attention supposedly provides for non-linear tasks, and the feasibility of extending these findings from non-language to language models?**
>
> **Reply:** For the first part of your query about linear and non-linear attention, please refer to our response to Q2. For the second part, we believe that a key challenge in proving in-context learning results for language problems lies in understanding how the structure of a given grammar aligns with the expressive power of self-attention. Specifically, this involves questions such as how compactly a verifier can represent a proof that a sentence belongs to a particular grammar.
>
>
> **Q3:implications for practitioners from your work?**
>
> **Reply:** This is certainly an important question that we are actively thinking about. There are few takeaways that may be of interest to practitioners
> Depth matters: For Transformers to solve a diverse set of ICL problems (which language models do), depth plays a very important role. This could be a valuable insight for architecture design, if the focus is ICL abilities.
> Non-robustness: It was interesting to learn from the theory that multilayer models, if too deep relative to training task diversity, can learn non-robust solutions. It does so by using different layers to solve different tasks seen during training. Thus if we are going deeper, we better make sure to have a more diverse training dataset.
> Looped models: One fascinating finding is that looped models, although seemingly restrictive, can solve ICL with close to optimal depth in many cases. Furthermore they afford significantly higher robustness since they are forced to learn a more generalizable algorithmic solution, rather than the “shortcuts” mentioned above. This suggests that looped models should be given more attention if such algorithmic abilities are of interest

---

### Author Response · Authors · 2024-11-26
**General response**

****We thank the reviewers for their valuable feedback to improve our work. We deliberately applied all of the comments to the paper, and uploaded the new version. We have left the changes compared to the previous version in blue. We kindly ask the reviewers to reconsider their score based on the improved version that we uploaded.****

---

### Meta-Review · Area_Chair_Y5zq · 2024-12-24

**Metareview:**

This paper considers learning ability of Transformers to perform in-context learning where the covariance of the input can differ in each task. It is shown that Transformers can implement multi-step gradient descent. In addition to that, the authors discuss robustness of Transformers, that is, the multi-layer Transformers are much less robust than Looped Transformers with parameter sharing.

This paper somehow lacks its novelty. The presented upper-bound is already indicated by existing work. The novelty mainly lies in the lower bound analysis. However, the lower bound is not really about Transformers but its proof relies on a polynomial argument.
In addition to that, the proof is not well organized. The logic in the proofs is not clearly described. It requires substantial revision.

As a summary, the results in this paper are not so strong and the writing should go through substantial revision. Thus, I do not recommend acceptance of this paper.

**Additional Comments On Reviewer Discussion:**

Some technical questions raised by the reviewers were properly addressed by the authors. However, the essential weakness of this paper could not be resolved by a minor revision.

---

### Decision · Program_Chairs · 2025-01-22

Reject